# Does curve pattern impact on the effects of physiotherapeutic scoliosis specific exercises on Cobb angles of participants with adolescent idiopathic scoliosis: A prospective clinical trial with two years follow-up

Yunli Fan[1,2,3], Michael K. T. To[1,2], Eric H. K. Yeung[3], Jianbin Wu[1], Rong He[3], Zhuoman Xu[3], Ruiwen Zhang[3], Guangshuo Li[3], Kenneth M. C. Cheung[1,2], Jason P. Y. Cheung[1,2]*

1 Department of Orthopaedics, The University of Hong Kong – Shenzhen Hospital, Shenzhen, Guangdong Province, People's Republic of China, 2 Department of Orthopaedics and Traumatology, The University of Hong Kong, Hong Kong SAR, China, 3 Department of Physiotherapy, The University of Hong Kong – Shenzhen Hospital, Shenzhen, Guangdong Province, People's Republic of China

* cheungjp@hku.hk

## Abstract

### Background

Current clinical evidence suggests that a well-planned physiotherapeutic scoliosis specific exercise (PSSE) program is effective for scoliosis regression.

### Objectives

We investigated the effect of curve patterns on Cobb angles with PSSE.

### Methods

This was a non-randomized prospective clinical trial that recruited participants with adolescent idiopathic scoliosis between January and June 2017. Participants were grouped by curve pattern into major thoracic and major lumbar groups. An outpatient-based PSSE program was conducted with the following schedule of intensive exercise: $\geq$ 1 session of supervised PSSE per month and > 30min of home exercise 5 days/week in the first 6 months, after which exercise frequency was reduced to 1 session of supervised PSSE every three months and > 30min of home exercise 5 days/week until 2 years after study initiation. Radiographic Cobb angle progressions were identified at the 1, 1.5 and 2-year follow-ups. A mixed model analysis of variance (ANOVA) was performed to examine the differences in Cobb angles between groups at four testing time points. The two-tailed significance level was set to 0.05.

**Data Availability Statement:** All relevant data are within the manuscript and its Supporting information files.

**Funding:** Funding Project: Sanming Project of Medicine (SZSM201612055) "Team of Excellence in Spinal Deformities and Spinal Degeneration Diseases" in Shenzhen, Guangdong province, China. The University of Hong Kong - Shenzhen Hospital Seeding Project, HKUSZH201902042, Jason PY Cheung Receiver: Kenneth MC Cheung, Jason PY Cheung.

**Competing interests:** The authors have declared that no competing interests exist.

## Results

In total, 40 participants were recruited, including 22 with major thoracic curves (5 males and 17 females; mean age 13.5±1.8 years; Cobb angle 18–45 degrees) and 18 with major lumbar curves (7 males and 11 females; mean age 12.7±1.7 years; Cobb angle 15–48 degrees). Curve regressions, namely the reduction of Cobb angles between 7 to 10 degrees were noted in 9.1% of participants in the major thoracic group; reductions of 6 to 13 degrees were noted in 33.3% of participants in the major lumbar group at the 2-year follow-up. Repeated measurements revealed a significant time effect ($F_{2.2,79.8} = 4.1$, $p = 0.02$), but no group ($F_{2.2,79.8} = 2.3$, $p = 0.1$) or time × group ($F_{1,37} = 0.97$, $p = 0.3$) effects in reducing Cobb angles after 2 years of PSSE. A logistic regression analysis revealed that no correlation was observed between curve pattern and curve regression or stabilization (OR: 0.2, 95% CI: 0.31–1.1, $p = 0.068$) at the 2-year follow-up.

## Conclusion

This was the first study to investigate the long-term effects of PSSE in reducing Cobb angles on the basis of major curve location. No significant differences in correction were observed between major thoracic and major lumbar curves. A regression effect and no curve deterioration were noted in both groups at the 2-year follow-up.

## Trial registration

ChiCTR1900028073.

## Introduction

Adolescent idiopathic scoliosis (AIS) is a three-dimensional spinal deformity with an unknown etiology, characterized by lateral deviation in the frontal plane, axial rotation in the horizontal plane and an abnormal sagittal curvature [1]. Surgery is typically recommended if a spinal curve reaches 50 degrees, because such curves are associated with a continued progression risk into adulthood [1–3]. Surgical fusion causes spinal stiffness and should be avoided if possible [3]. Thus, the goal of conservative managements including bracing and exercises, is to prevent spinal deformity deterioration past the operative threshold [4].

Options for nonsurgical management vary widely and depend on the prognostic evaluation of curve progression [4]. Curve type is an established risk factors of scoliosis progression [5]. In particular, a thoracic curve with a larger Cobb angle has a higher likehood of progression than single lumbar or thoracolumbar curves do [2]. Therefore, understanding the effect of curvature on interventions can help clinicians select appropriate treatments for patients.

Bracing is commonly prescribed for moderate scoliosis, and its corrective effect is influenced by the location of structural curves [6, 7]. In addition, scoliosis-specific exercise is a nonoperative option that is well received by patients and families [8]. Several systematic reviews and randomized controlled trials have reported the positive effects of physiotherapeutic scoliosis specific exercise (PSSE) on slowing curve progression as well as improving cosmetic and quality of life [9–12]; however, the relationship of curve location with correction effects has not been clearly discussed in these studies, in which small samples of participants have been recruited and followed up over the short-term [13–15]. In particular, one 6-month long

randomized control trial (RCT) revealed a significant correlation between thoracic curvature with an imbalanced pelvis (n = 15) and the largest curves after Schroth treatment [13]; one 4-month long RCT demonstrated the greater Cobb reduction was noted in the thoracic region (n = 20) after body awareness exercise [14]. Therefore, a clinical trial, in which curve pattern is controlled, is warranted to investigate the long-term effects of curve pattern and PSSE on reducing Cobb angles. This is valuable for clinical practice because the deterioration is possible after initial improvement in short-term follow-up.

According to current evidence from PSSE studies [9, 10, 16], the Schroth is the most used PSSE approach. It adopts a specific respiratory technique, asymmetrical breathing in the diagonal direction, to achieve vertebral and rib cage derotation [17]. It utilizes muscle activation and emphasizes core muscle stabilization in a corrected posture throughout the day to change habitual postures and improve spinal alignment [17, 18]. However, the influence of curve magnitude on Schroth method outcomes is unclear. The spinal deformity profile that leads to optimal outcomes with Schroth exercise is also unclear. Thus, this study investigated the long-term therapeutic effects of Schroth exercises on major thoracic and major lumbar curves, respectively. We hypothesized that patients with lumbar curves, which have a lower reported risk of curve progression and higher spinal segmental flexibility, benefit more from long-term PSSE treatment than do patients with major thoracic curves. The current results can provide physiotherapists with additional information to develop individualized exercises based on curve magnitude. These results can also form the basis for further RCTs evaluating the relationship between curve patterns and PSSEs.

## Materials and methods

### Study design

This was a non-randomized prospective clinical trial in which participants with AIS were recruited consecutively between January and June 2017 from one scoliosis clinic at the University of Hong Kong—Shenzhen Hospital (HKU—SZH), China. This study obtained ethical approval by the institutional review board of the HKU—SZH with reference number: GNSF201603 (S1 File). Individuals in this manuscript have given written informed consent to publish these case details. The authors confirm that all ongoing and related trials for this intervention are registered in the Chinese clinical trial registry (www.chictr.org.cn, trial registration: ChiCTR1900028073) (S1 File). Participants and legal guardians signed a written informed consent for PSSE treatment and participants were followed up for 2 years thereafter. The inclusion criteria were age 10 to 16 years; bone immaturity in terms of Risser sign was less than 5 [19]; and Cobb angles [20] from 10 degrees to 50 degrees. Exclusion criteria were diagnoses other than AIS; disabilities or systemic illnesses that prevent participants from performing exercise; patients being unable to attend one session per month of supervised PSSE during the first 6 months, this was to review performed exercise monthly to control learning effects; hypermobility (Beighton score [21] greater than 4) and previous treatment for AIS. Joint hypermobility (JHM) is reported to have a higher prevalence in patients with single curves than in those with double-curve scoliosis; and this may affect physiotherapy outcomes [22]. Therefore, patients with hypermobility were excluded from this study to control for systematic error introduced by JHM differences between curve patterns. Participants were categorized into two groups according to their whole spine radiographic characteristics [23, 24]: participants with 3C (major thoracic with or without a minor lumbar curve) or N3N4 curves (double, well balanced curves) formed the major thoracic group; and those with single lumbar, single thoracolumbar, or 4C curves (a major lumbar curve with a minor thoracic) comprised the major lumbar group (Fig 1).

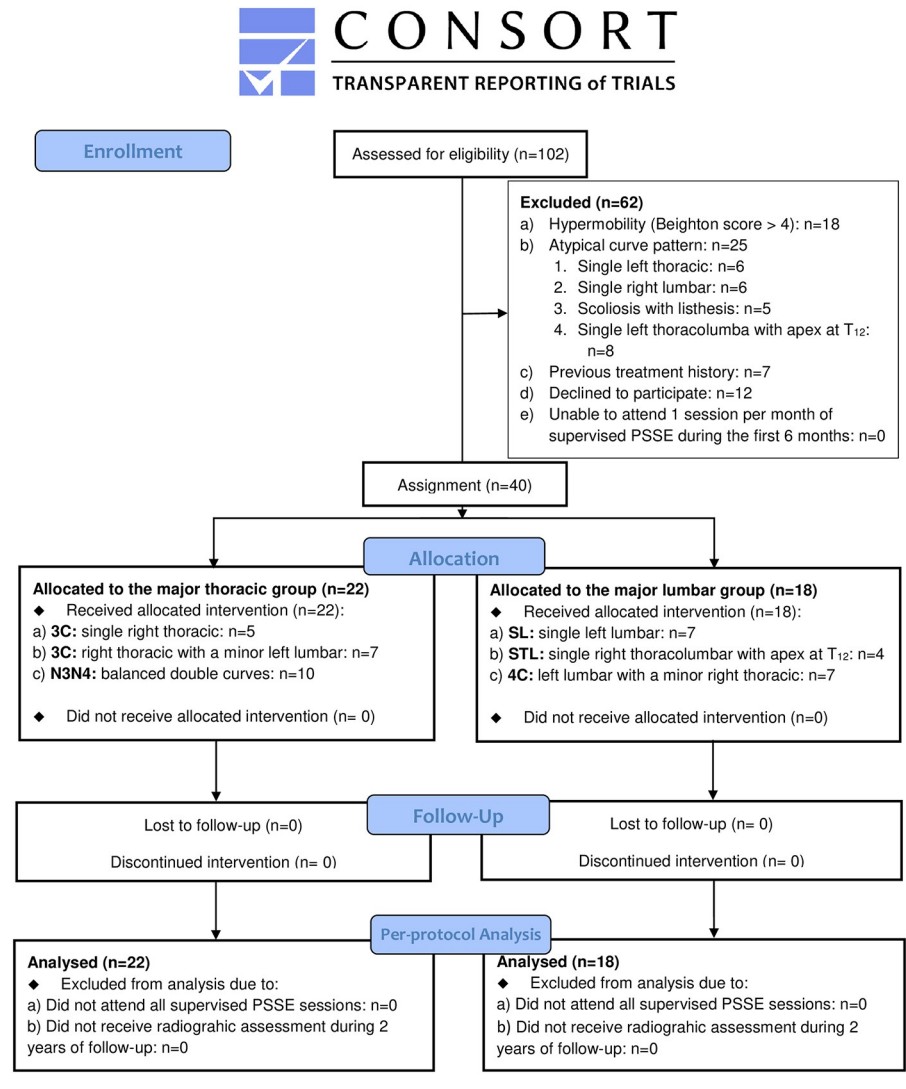

**Fig 1. CONSORT flow diagram.**

## Study intervention

All participants participated in a 6-month intensive PSSE program supervised by two PSSE-Schroth certified physiotherapists (Barcelona Scoliosis Physical Therapy School [BSPTS] approach) [23]. Supervised treatment during the 6 months occurred $\geq$1 time per month for 1 hour each time. Participants were instructed to perform >30min sessions of home exercise at least $\geq$5 times per week. For participants who met Scoliosis Research Society bracing criteria, Cheneau bracing was prescribed prior to the PSSE program [25]; after an adaption period (2 weeks) of wearing a brace, an initial in-brace angle [26–28] was measured by a spine surgeon who was blinded to this study, using the X-ray with a brace fitted on the participant more than two hours [28]. Bracing compliance was reported by patients on their monthly treatment adherence checklist (S2 File). After the first 6 months of PSSE, all participants reduced their frequency of supervised PSSE to once every 3 months to review their exercises, which they

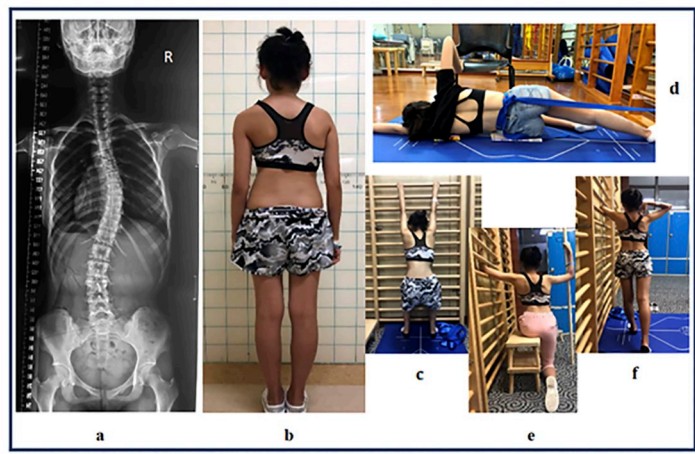

**Fig 2. Exercises demonstration for major right thoracic curves. a:** a major right thoracic (T4 to 12) with a Cobb angle of 30 degrees. **b:** posterior posture view of the participant. **c:** short semihanging with caudal-cranial spine lengthening. **d:** corrective exercise in side-lying with shoulder counter traction. **e:** corrective exercise in sitting with pelvic pulling strategy on the side of weak point and shoulder counter traction. **f:** corrective exercise in standing with pelvic derotation and shoulder counter traction.

performed for 2 years after completing the PSSE program. During this time, the frequency of home exercise was maintained. Attendance for supervised PSSE in an outpatient setting was recorded in the hospital's prospective database system. Participants were requested to document home exercise and bracing compliance on daily basis using a treatment adherence checklist (S2 File), and handed it to a physiotherapist assistant who was blinded to this study every month during the first 6-month study period; then they were required to submit the checklist every 3 months thereafter. Exercise compliance was calculated in hours per week per year for analysis. The PSSE intervention consisted of 50 breaths per exercise. Short semihanging with caudal-cranial spine lengthening (Fig 2c), corrective exercise in side-lying with shoulder counter traction (Fig 2d), corrective exercise in sitting with pelvic pulling strategy on the side of the weak point and shoulder counter traction (Fig 2e), corrective exercise in standing with pelvic derotation and shoulder counter traction (Fig 2f) were prescribed for participants in the major thoracic group. Short semihanging with caudal-cranial spine lengthening (Fig 3c), corrective exercise in side-lying with passive correction on the lumbar prominence (Fig 3d), corrective exercise in standing with shoulder counter traction (Fig 3e), muscle cylinder in standing for opening lumbar concavity (Fig 3f) and the corrective exercise in sitting with shoulder counter traction (Fig 3g) were prescribed to participants in the major lumbar group. The difference between the two groups' exercises was that a muscle cylinder was prescribed only to participants with major lumbar curves. The muscle cylinder exercise in a kneeling position is typically prescribed to patients with major thoracic curves [17]; however, patients commonly complain that it causes knee pain. Therefore, considering the potential adverse effects during long-term follow-up, the muscle cylinder in a kneeling position was not prescribed in this study. Moreover, dynamic exercises, such as Schroth walking was not used in this study because of limited space in our center. In addition, participants were asked to record and report any discomfort, such as muscle fatigue and muscle sprain, to the on-duty physiotherapist during the study period.

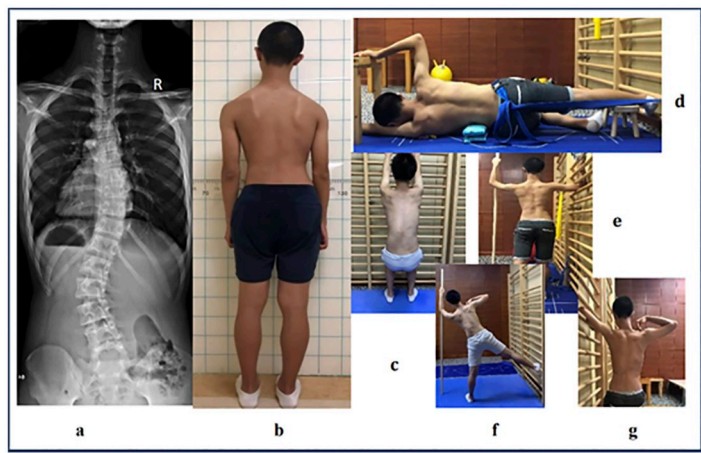

**Fig 3. Exercises demonstration for major left lumbar curves. a:** a major left lumbar curve (T12 to L4) with a Cobb angle of 48 degrees. **b:** posterior posture view of the participant. **c:** short semihanging with caudal-cranial spine lengthening. **d:** corrective exercise in side-lying with passive correction on the lumbar prominence. **e:** corrective exercise in standing with shoulder counter traction. **f:** muscle cylinder in standing for opening lumbar concavity. **g:** corrective exercise in sitting with shoulder counter traction.

## Outcome measurement

Participants underwent whole-spine radiography in a standing position with arms slightly abducted at the sides of body. The Cobb angles of major curves were assessed by a spine surgeon who was blinded to this study. Cobb angles were recorded in the initial assessment and every 6 months after the PSSE program until the 2-year follow-up. An in-brace Cobb angle was measured using a radiography with the brace fitted on the participant. A percentage value of the initial in-brace correction ($IBC = \frac{Cobb\_in-brace}{Cobb\_baseline}$) was compared between groups to evaluate the bracing effect [29]. Participants remained out of their braces overnight prior to obtaining radiography at the two years of follow-up [28]. Curve regression was defined as a decrease in the Cobb angle by 6 degrees or more, stabilization was defined as a <6 degrees change in Cobb angle, and deterioration was defined as a Cobb angle increase of 6 degrees or more [2, 30].

## Statistics

**Sample size estimation.** To detect an effect size of 0.50, 80% power for a two-tailed test, and a correlation of 0.6 for repeated measures of Cobb angles over 4 time points, 13 participants were required for each group [31]. Considering a possible 35% dropout rate during the 2-year study period, each group were planned to have 18 participants (Fig 1).

**Statistical analysis.** Descriptive statistics with 95% Confidence Interval (CI) were calculated for baseline variables including demographics, radiography, exercise compliances and Lonstein and Carlson Risk of progression values ($LCR - value = \frac{Cobb-3\times Risser}{Age}$) [32] for all participants (Table 1) [33]. Changes in Cobb angles, in the form of the difference value [D-value = ($Cobb_{6-month/1-/1.5-/2-year}$)-($Cobb_{baseline}$)] were measured to detect curve progression (D-value $\geq$ 6 degrees), stabilization (D-value < 6 degrees, or D-value > -6 degrees) and regression (D-value $\leq$ -6 degrees) between groups [34]; raw Cobb angles were repeatedly compared over four testing time points to detect time, group and time $\times$ group interaction effects in reducing Cobb angles. Intention to treat and per-protocol analyses would be performed based on

**Table 1. Baseline characteristics of participants.**

| | Major thoracic group (95% confidence interval) | Major Lumbar group (95% confidence interval) |
|---|---|---|
| **Age** (years: Mean ± SD) | 13.5 ± 1.8 (12.8–14.6) | 12.7 ± 1.7 (12.6–13.7) |
| **Sex** | | |
| Female | 17 (77.3%) | 11 (61.1%) |
| Male | 5 (22.7%) | 7 (38.9%) |
| **Risser sign** | | |
| 0–2 / 3–4 | 9 / 13 | 9 / 9 |
| **Initial Cobb Angle** (min-max) | 18–45 | 15–48 |
| (degrees: Mean ± SD) | 28.5 ± 8.4 (24.5–32.5) | 26.8 ± 10.1 (22.4–31.2) |
| **Curve pattern** | | |
| 3C / SL or STL | 12 | 11 |
| N3N4 / 4C | 10 | 7 |
| **Braced** (yes) | 7 | 3* |
| In-brace hours (hours/day) | 13.6 ± 6.9 | 16.7 ± 7.5* |
| full-time brace ($\geq$ 21hr/day): n | 3 | 2 |
| night-time brace ($\leq$ 8hr/day): n | 4 | 1* |
| **Initial in-brace Cobb angle** | | |
| (degrees: Mean ± SD) | | |
| **Initial in-brace correction rate (%)** | 23 ± 4.4 | 26 ± 6 |
| | 37 | 37 |
| **Initial progression risk** | | |
| Lonstein and Carlson Risk of progression | 44% | 46% |
| **Exercise compliance** | | |
| 1st year: hours/week | 5.5 ± 2.2 | 5.5 ± 2.1 |
| 2nd year: hours/week | 3.0 ± 0.8$^{§ ¶}$ | 4.1± 2.1 |

**3C**: A single right thoracic curve or a major right thoracic with a minor left lumbar. **SL**: A single left lumbar. **STL**: A single right thoracolumbar. N3N4: double curves but well balanced. **4C**: A major left lumbar with a minor right thoracic.

* two participants refused brace treatment due to cosmetic concerns which introduced an imbalance in bracing between groups.

§ An independent *t-test* revealed a significant difference between groups.

¶ A paired *t-test* revealed a significant difference in the major thoracic group.

participants' original allocation if any dropouts or crossovers were observed. However, only per-protocol analysis was adopted in this study because no crossovers or dropouts were noted during the whole study period (Fig 1). A mixed model analysis of variance (ANOVA) was conducted if the normality assumptions of Cobb angles were met whereas a nonparametric Friedman-type test would be performed if normality was violated. Post hoc pairwise comparisons, in terms of repeated measures with the Bonferroni adjustments were performed if any significant differences were detected in either within-subjects or between-subjects comparisons. A Logistic regression was performed to detect correlations of brace treatment and curve pattern with the regression, stabilization or deterioration after 2 years of PSSE. The data were analyzed using SPSS version 20.0 (IBM, Chicago, IL). The level of significance was set to 0.05 with a two-tailed test.

## Results

### Participants

In total, 102 patients were screened to confirm eligibility, after which 62 patients were excluded for the following reasons: hypermobility (Beighton score > 4) was detected in 18 patients (single right thoracic: n = 7, single left lumbar: n = 5; right thoracic with a minor lumbar: n = 6); 25 patients had reported, atypical curve patterns [35–38] (single left thoracic: n = 6 [36], single right lumbar: n = 6 [38], scoliosis presented with vertebral listhesis: n = 5 [37], single left thoracolumbar with apex at $T_{12}$: n = 8 [35]); 7 patients were previously treated with either bracing or exercise; 12 patients declined to participate in this study. Therefore, 40 participants were recruited and all completed follow-up (Fig 1). The major thoracic group had 22 participants (5 males and 17 females). Of them, 12 had 3C curves (single right thoracic: n = 5; right thoracic with a minor left lumbar: n = 7) and 10 had N3N4 curves (balanced right thoracic with a left lumbar). The major lumbar group had 18 participants (7 males and 11 females). Of them, 7 had single left lumbar curves, 4 had right thoracolumbar curves with apex at $T_{12}$ and 7 had 4C curves (left lumbar with a minor right thoracic). The mean age of participants in the major thoracic and major lumbar groups was 13.5±1.8 years and 12.7±1.7 years, respectively (Table 1). At baseline, the two groups did not differ statistically in age, sex, Risser sign (0–2 / 3–4), proportion of sub-curvature types, LCR-value, or initial Cobb angle of the major curve (Table 1). Ten participants with brace treatment who all started Cheneau orthosis [39] before commencing the PSSE and achieved 37% of initial in-brace correction [27] in each group (Table 1). In particular, 5 of them were wearing a night-time brace only (≤8 hours/day) whereas 5 were wearing a full-time brace (≥21 hours/day) (Table 1). Two participants in the major lumbar group refused brace treatment due to cosmetic concerns, which introduced an imbalance in bracing between groups. Thus, bracing was set as a covariate with the repeated analysis to evaluate effects of PSSE in reducing Cobb angles.

### Exercise compliance

The two groups differed significantly in exercise compliance during the second year of follow-up (independent *t-test*, p = 0.04) (Table 1). In addition, mean exercise compliance of the major thoracic group was significantly decreased in the second year (paired *t-test*, p<0.01) (Table 1); the mean exercise compliance of the major lumbar group was 5.5±2.1 hours/week in the first year and slightly decreased to 4.1±2.1 hours/week in the second year (Table 1). There were no adverse effects detected during this study period.

### Intervention effects

In total, 23 participants (57.5%) with Cobb angle < 30 degrees refused to repeat radiography at the 6-month follow-up after the PSSE program because of radiation exposure concerns; therefore, no comparison of participants before and at the 6-month follow-up was performed. Nonetheless, all participants completed radiography at the 1-year, 1.5-year and 2-year follow-up, respectively. Full set of raw data is available in the S3 File.

A per-protocol analysis was performed to analyze the raw value of Cobb angles over four testing time-points between groups. Bracing (hours/day) was set as a covariate with a mixed model repeated analysis since an imbalance of bracing was noted at study initiation. Levene's test indicated that Cobb angles at the baseline met homogeneity of variance assumption (p = 0.5); Mauchly test indicated that the assumption of sphericity was violated ($\chi^2$ = 32.1, df = 5, p < 0.001); hence, the degrees of freedom were corrected using Greenhouse-Geisser estimates of sphericity [40]. With current sample size (n = 40) and by adjusting for bracing as

**Table 2. Per-protocol repeated measures of Cobb angles over four testing time points.**

| Testing timepoint | Group | Mean | Standard Deviation | 95% Confidence Interval | Minimum | Maximum | *P* |
|---|---|---|---|---|---|---|---|
| Baseline | T | 28.5 | 8.4 | 24.8–32.2 | 18 | 45 | - |
| | L | 26.8 | 10.1 | 21.8–31.8 | 15 | 48 | - |
| | Total | 27.7 | 9.1 | 25.6–29.9 | 16 | 46 | - |
| 1-year | T | 27 | 7.2 | 23.8–30.2 | 18 | 38 | 0.910 |
| | L | 24.7 | 9.4 | 20.0–29.4 | 10 | 42 | 0.476 |
| | Total | 26.0 | 8.2 | 23.8–28.0 | 14 | 40 | 0.107 |
| 1.5-year | T | 27.1 | 7.7 | 23.7–30.5 | 10 | 40 | 1.0 |
| | L | **23.3**[*] | 9.4 | 18.6–28.0 | 10 | 45 | **0.043** |
| | Total | **25.4**[t] | 8.6 | 23.2–27.3 | 10 | 43 | **0.015** |
| 2-year | T | 27.5 | 7.0 | 24.4–30.5 | 15 | 40 | 1.0 |
| | L | **22.9**[*] | 9.2 | 18.4–27.5 | 10 | 43 | **0.009** |
| | Total | **25.4**[t] | 8.2 | 23.4–27.1 | 13 | 42 | **0.005** |

A Bonferroni adjustment was performed with repeated measures to control the familywise Type I error. **Group T:** major thoracic group. **Group L:** major lumbar group.

[*] significantly differ from the baseline in the major lumbar group.

[t] significantly differ from the mean value at baseline for all participants.

a covariate, the repeated analysis showed a significant within-subjects time effect ($F_{2.2,79.8} = 4.1$, $p = 0.02$), but no group ($F_{2.2,79.8} = 2.3$, $p = 0.1$), bracing ($F_{2.2,79.8} = 1.4$, $p = 0.3$) or time × group ($F_{1,37} = 0.97$, $p = 0.3$) effects were detected in reducing Cobb angles after 2 years of treatment (Table 2). A post hoc pairwise comparison (within-subjects) with a Bonferroni adjustment revealed a significant reduction of Cobb angle, regardless of curve pattern, was observed at the 1.5-year (27.7±9.1 degrees vs 25.4±8.6 degrees, $p = 0.015$) and the 2-year (27.7 ±9.1 ±9.1 degrees vs 25.4±8.2 degrees, $p<0.01$) follow-up (Table 2). A separate repeated analysis with a Bonferroni adjustment further suggested that the significant reduction of Cobb angle was observed in the major lumbar group ($F_{3,51} = 6.2$, $p<0.01$), in which a reduction of Cobb angle was noted after training for one and a half years (26.8±10.1 degrees vs 23.3±9.4 degrees, $p = 0.04$) with a maintained reduction after two years of follow-up (26.8±10.1 degrees vs 22.9±9.2 degrees, $p<0.01$) (Table 2).

Overall, we observed 20% with regression (n = 8) and 80% with stabilization in this study, in which 65% of participants (n = 26) reached bone maturity (Risser: 5) after two years of PSSE and no curve pattern was changed (Table 3). A chi-square analysis revealed that there was no

**Table 3. Descriptive statistics of demographic and radiographic variables at the 2-year follow-up.**

| The 2-year follow-up | | Thoracic curves (n = 22) | Lumbar curves (n = 18) |
|---|---|---|---|
| **Age** (years: Mean ± SD) | | 15.6 ± 1.8 | 14.7 ± 1.7 |
| **Risser sign** 0–2,3–4,5 (n) | | 0,4,18 | 0,10,8 |
| **Cobb angle** (degrees: Mean ± SD) | | 27.5 ± 7.0 | 22.9 ± 9.2 |
| $\geq$ 30 degrees: n (%): min—max | | 8 (36.4%): 31˚–40˚ | 5 (27.8%): 30˚–43˚ |
| <30 degrees: n (%): min—max | | 14 (63.6%):15˚- 28˚ | 13 (72.2%): 10˚ -27˚ |
| a) **Regressed**: n (%): min to max of D-value | | 2 (9.1%): -6˚ to -10˚ | 6 (33.3%): -6˚ to -13˚ |
| b) **Stabilized**: n (%): min to max of D-value | | 22 (90.9%): 5˚ to -5˚ | 12 (66.7%): 3˚ to -5˚ |
| c) **Progressed**: n (%): min to max of D-value | | 0 | 0 |
| **Curve pattern:** | 3C / SL or STL (n) | 12 | 11 |
| | N3N4 / 4C (n) | 10 | 7 |

difference ($\chi^2$ = 3.6, df = 1, $p$ = 0.057) in distributions of curve regression and stabilization at the 2nd year follow-up between groups. No deterioration of the major curvature occurred in either group (both average and individual data) at the 2-year follow-up. A logistic regression analysis further revealed that there was no correlation of bracing (OR: 0.69, 95% CI: 0.10–4.82, $p$ = 0.71) or curve patterns (OR: 0.2, 95% CI: 0.31–1.1, $p$ = 0.068) with curve regression or stabilization at the 2-year follow-up.

In the major thoracic group, descriptive statistics revealed that curve regressed (Cobb angles: -8.8±2.5 degrees) in 18.2% of participants and stabilized (Cobb angle: -2.9±1.6 degrees) in 81.8%, none deteriorated in the first year. Only 2 participants maintained curve regression in the second year, which resulted in 9.1% of participants showing regression (Cobb angles: -8.5±2.1 degrees) (Table 3). Thus, up to 90.9% of participants stabilized (Cobb angles: -3.1±1.6 degrees), and no deterioration was detected (Table 3).

In the major lumbar group, descriptive statistics revealed that curves regressed (Cobb angles: -9.5±4.9 degrees) in 11.1%, stabilized (Cobb angles: -3.4±1.2 degrees) in 83.3% and deteriorated by 7 degrees in one subject after the first year of training. In the second year of follow-up, up to 33.3% of participants had improved Cobb angles (Cobb angles: -8.2±2.5 degrees), while 66.7% stabilized (Cobb angles: -3.2±1.6 degrees), and no deterioration was noted (Table 3). Of the participants with curve regression in the first year, one maintained curve regression with a reduction of Cobb angle in 13 degrees, and one only gained an additional degree in the second year. Moreover, one participant with a single left lumbar curve of 31 degrees (Fig 4a) who refused brace treatment obtained a Cobb angle of 23 degrees (Fig 4b) after one and a half years of PSSE. In addition, another participant with a single right thoracolumbar curve of 44 degrees who refused brace treatment (Fig 5a) obtained a Cobb angle of 31 degrees (Fig 5b) after two years of PSSE.

## Discussion

This was the first study to investigate the long-term outcomes of patients with different AIS curve patterns undergoing PSSE treatment. The findings of this study indicate no significant difference in Cobb angle changes between groups after regular PSSE for two years, regardless of the curve pattern. This implies that the curve pattern does not significantly impact the effects of PSSE in participants with AIS. A previous 6-month long study suggested that the curve pattern, a thoracic curvature with an imbalanced pelvis (n = 15), is significantly correlated with the largest sum of curves after Schroth exercise, and that study further explained this might be due to a higher risk of progression in thoracic scoliosis [13]. However, with a short-term follow-up and limited sample size, that study could not draw any firm conclusions on the relationship of curve pattern and Schroth exercise in treating AIS. Our results extend this discussion with a longer follow-up period, and curve patterns were controlled before PSSE commenced. In this study, no curve progression was observed in either group, suggesting that a well-planned PSSE program can prevent scoliosis deterioration regardless of curve patterns. This result is consistent with previous findings that PSSE reduces curve progression in patients with mild curve [41–43]. Compared with those studies, we observed regression effects in larger curves and with a longer follow-up period. Moreover, participants in our study presented a higher progression risk (Table 1: LCR-value: 44% vs 46%) at the study initiation; additionally, participants with brace treatment and all received the same initial in-brace correction of 37% improvement. The lack of deterioration may be attributable to the effects of both bracing and PSSE; however, because comparable initial in-brace correction explained the effects of bracing, the effects from the PSSE could be highlighted in our study. Therefore, our results revealed the effectiveness of PSSE programs in curve stabilization and regression of scoliosis for patients

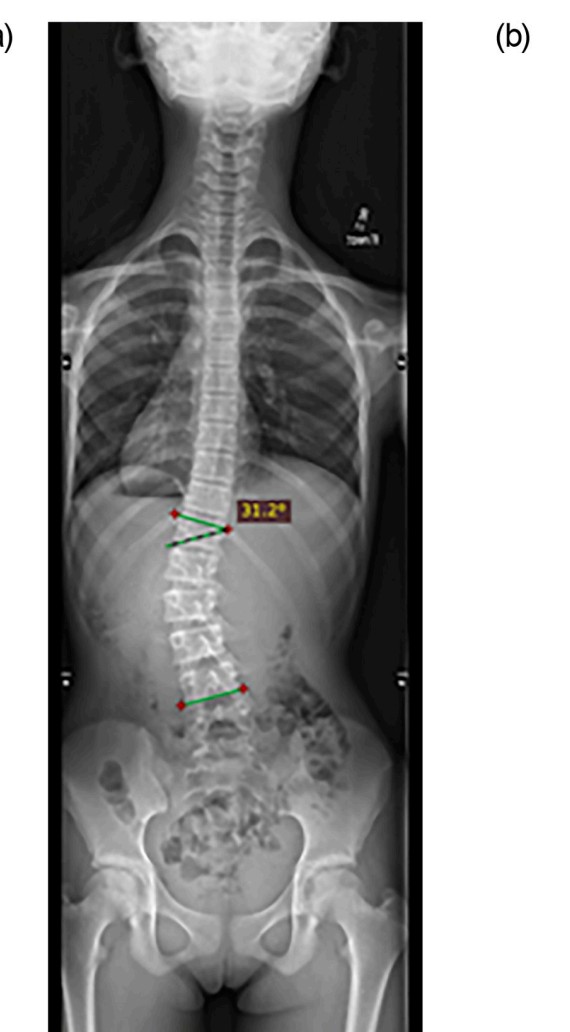
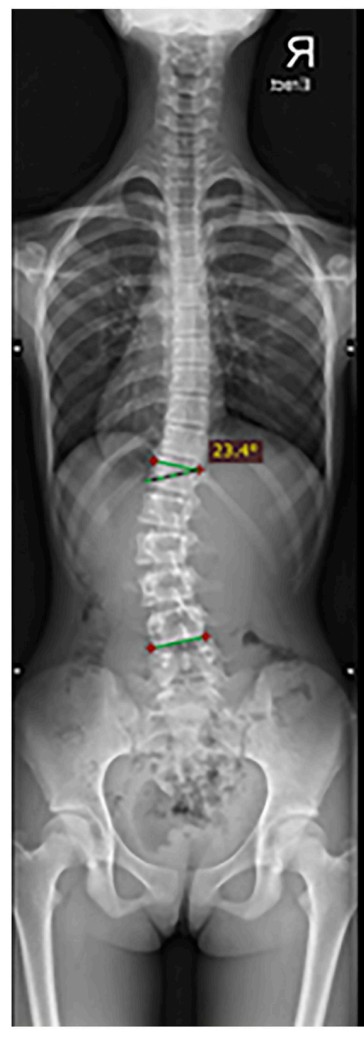

**Fig 4. Radiographic assessment (posterior anterior view) of one participant with a single left lumbar scoliosis who refused brace treatment for cosmetic reasons. a:** an initial Cobb angle of 31 degrees from T12 to L4. **b:** a Cobb angle of 23 degrees from T12 to L4 at the 1.5-year of follow-up.

with mild to moderate scoliosis. However, a risk of overtreatment exists because patients with only mild curves may not experience progression. Therefore, the burden of treatment should be avoided as much as possible through periodical assessments [13]. Optimal time points for assessment require future study.

After two years of follow-up, no clinical deterioration was detected in either group, indicating a promising effect of PSSE on the prevention of scoliosis progression. Moreover, the regression effect accompanied by a notable decease of progression risk after 2 years. In particular, 65% of participants reached skeletal maturity (Risser 5) and 68% of participants presented <30 degrees of Cobb angle after two years of treatment (Table 3). This is promising because of curve less than 30 degrees are unlikely to progress after skeletal maturity [2]. In particular, a decrease of exercise compliance at the second year may have led to less of a regression effect at the 2-year follow-up in the major thoracic group; a significant time effect in reduction of Cobb angles was only noted for participants with the major lumbar curves; moreover, a difference was observed as early as one and a half years after training, and reductions were maintained at

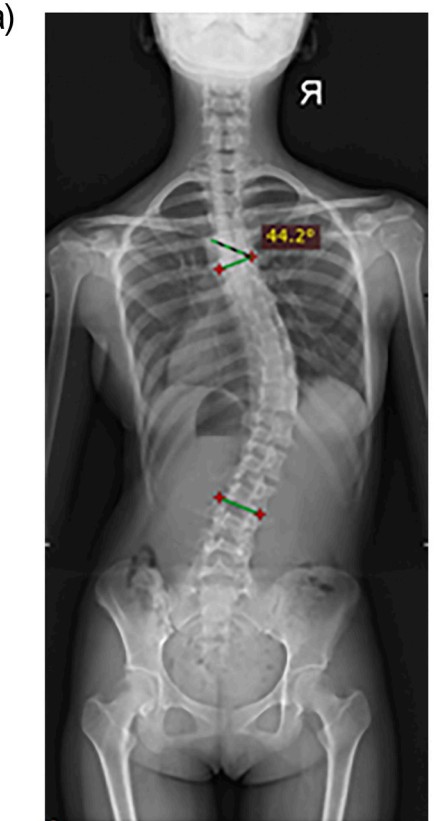
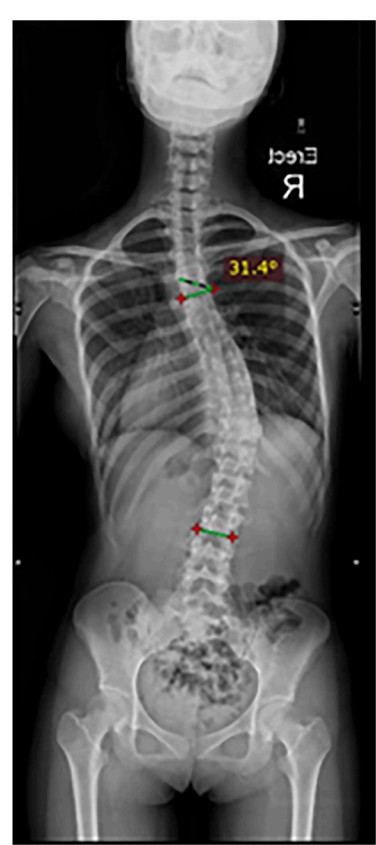

**Fig 5. Radiographic assessment (posterior anterior view) of one participant with a single right thoracolumbar scoliosis who refused brace treatment for cosmetic reasons. a:** an initial Cobb angle of 44 degrees from T5 to L2. **b:** a Cobb angle of 31 degrees from T5 to L2 at the two-year follow-up.

the end of the study period. This might be parallel with our original study hypothesis that major lumbar curves benefited more from long-term exercise in terms of having a lower progression risk. However, this should be interpreted with caution because progression risk estimates, in terms of LCR-values were 44% for thoracic curves and 46% for lumbar curves, LCR values were comparable between groups at the study initiation. Hence, this observed promising result with lumbar curves might be explained by the assumed more important flexibility of the lumbar spine than the thoracic spine, confounded with a better exercise compliance and its associated effects of good core lumbar muscle stabilization. The superior flexibility of the lumbar spine compared to the thoracic spine has been proven in biomechanical studies [44–46]. The lumbar spine is mainly stabilized by core muscles that are flexible, stretchable and respond to external forces very well, whereas the stabilization mechanism in the thoracic spine is reliant on the rib cage, which contributes to rigidity and stability [45–47]. Additionally, according to the general principles of the Schroth method, corrective strategies including side shift and axial elongation are done in both thoracic and lumbar regions, whereas spiral breathing is specifically adopted in the thoracic region [17]. Spiral breathing is difficult to learn and hard to control for most participants with AIS due to the requirement of breathing diagonally and asymmetrically not only in the frontal plane but also in the horizontal plane. Therefore, it is relatively easier for participants to correct the lumbar curvature via pelvic positioning, lengthening the curve concavity and stabilizing it in the proper position. In addition, stabilization

effects from core muscles are crucial for maintaining correction in lumbar spines [17, 47]. Therefore, the lumbar curvature may respond better to the three-dimensional correction forces from a musculoskeletal functional perspective; however, real-time feedback of the spine during different corrective exercises requires future study.

In our study, significant differences between groups in exercise compliance were noted in the second year. Higher compliance in the major lumbar curve might contribute to the larger Cobb angle reduction. This extends the results of some previous long-term studies of mild scoliosis that reported the only exercise intensity without reporting compliance [16, 48]. These previous studies have not addressed the relationship of compliance and treatment effects, which is of clinical value to define exercise protocols for subjects with PSSE treatment. In addition, our study revealed that an outpatient home-based PSSE program is a feasible protocol with reasonable compliance. The parents were asked to send monthly videos documenting the exercises performed to the Schroth therapist, which provided a reasonable variation control with respect to exercise quality and compliance. The frequency of such home-based programs was collected via a treatment adherence checklist, and exercise compliance was interpreted by how long participants spent on the PSSE program. This study calculated the supervised PSSE sessions and home exercise sessions in a sum number, in the form of the total hours per week. Hence, we were unable to analyze specifically whether supervised PSSE is superior to home exercises in curve correction despite previous accounts of such a relationship [49–52]. Besides, the bracing compliance was self-reported, and this might introduce recall effects, which can be improved if using a pressure sensor to monitor bracing hours. Nonetheless, only 10 participants wore a brace with 5 of which were wearing nighttime brace only, hence, bracing would not cause a big distraction in our study.

This study had a few inevitable limitations in addition to the small sample size. First, no untreated control group was included for comparison of intervention effects. However, because of the proven benefits of PSSE on AIS [11–13], it was unethical to leave participants without treatment for as long as 2 years. Thus, the benefit observed may have been influenced by natural recovery of the condition. However, the progression risk at study initiation was evaluated with approximately 45% of progression rate, and with a moderate Cobb angles of 29 degrees with major thoracic and 27 degrees with major lumbar curves, hence, the natural recovery was unlikely in our study. Second, we only enrolled participants representing 5 different types from 8 in the BSPTS classification [24]. The BSPTS concept categorizes scoliosis into 3C (A1, A2 and A3), 4C (B1 and B2), N3N4, single lumbar/thoracolumbar and double thoracic [23, 24]. For participants having a double curve type such as N3N4 or 4C, the minor curve will cause limits to overstretching and twisting of the concavity from the major curve that weaken the effects of correction in the major curve. However, the proportion of curves pattern in this study was similar (Table 1: 40.9% vs 44.4%) between the two groups, which could not cause large variations in the data. Besides, there were only 6 participants with single right thoracic curve and no double upper thoracic curves were presented in this study, therefore, it would decrease the observed power if we extra analysed the single thoracic curves independently. However, a comparison of the effects of PSSE for all eight curve types separately remains valuable for further study. Third, we were unable to blind the participants or physiotherapists due to the nature of PSSE studies, the exercise was prescribed and performed based on the understanding of the curve pattern by both therapists and patients, which may introduce methodological bias but also limited systematic errors such as learning effects. Performing exercise precisely is crucial to reach and maintain treatment effect, therefore, blinding of subjects and therapists was commonly sacrificed in most PSSE studies [10, 12]. However, the frequency and dosage of both supervised and home programs were strictly fixed, and we blinded the result assessor which controlled bias in the statistical analysis. Finally, we assumed the lumbar spine

responded more to PSSE because of its lower progression rate and with proven better segmental spinal flexibility than the thoracic spine. However, because of radiation exposure concerns, no bending radiographic assessment conducted to quantify spinal flexibility before the PSSE program. Thus, this study could not interpret the relationship between spinal flexibility and PSSE. Nonetheless, this study excluded participants with hypermobility that may cause systematic error during exercise. This controlled for potential heterogeneity when the spine started to respond to exercise. However, future study should address the relationship of spinal flexibility with PSSE, to define the correlation of curve location, segmental flexibility and PSSE in treating AIS. In addition to repeated measures of Cobb angles (raw value), we adopted a logistic regression analysis to investigate the correlation of curve pattern with scoliosis progression after 2 years of PSSE using a clinical significance threshold (D-value of a change in Cobb angle: $\geq 6$ degrees). This showed a value of clinical practice, but the analysis was underpowered regarding the small sample size [53]. Thus, a study with a bigger sample size is required to detect the correlation of curve pattern with scoliosis progression after PSSE treatment.

## Conclusion

PSSE treatment outcomes did not differ by thoracic or lumbar curve type. In general, PSSE with reasonable exercise compliance has positive effects on preventing scoliosis progression. However, further studies are necessary to address the correlation between spinal flexibility and the correction effects of PSSE when different curve magnitudes, curve types and levels are involved. The cosmetic and quality of life outcomes of PSSE should also be studied. This study formed the base for future larger scale studies and randomized trials.

## Supporting information

**S1 File. Study protocol.**
(PDF)

**S2 File. Treatment adherence checklist.**
(PDF)

**S3 File. Study data.**
(XLSX)

**S4 File.**
(RTF)

**S5 File.**
(PDF)

## Author Contributions

**Conceptualization:** Yunli Fan, Michael K. T. To, Rong He, Kenneth M. C. Cheung, Jason P. Y. Cheung.

**Data curation:** Jianbin Wu, Zhuoman Xu, Guangshuo Li.

**Formal analysis:** Yunli Fan.

**Funding acquisition:** Kenneth M. C. Cheung, Jason P. Y. Cheung.

**Investigation:** Eric H. K. Yeung, Ruiwen Zhang.

**Methodology:** Yunli Fan, Eric H. K. Yeung, Jason P. Y. Cheung.

**Project administration:** Eric H. K. Yeung, Kenneth M. C. Cheung, Jason P. Y. Cheung.

**Resources:** Jianbin Wu, Zhuoman Xu.

**Supervision:** Michael K. T. To, Jason P. Y. Cheung.

**Validation:** Yunli Fan, Kenneth M. C. Cheung, Jason P. Y. Cheung.

**Visualization:** Yunli Fan, Jason P. Y. Cheung.

**Writing – original draft:** Yunli Fan.

**Writing – review & editing:** Yunli Fan, Michael K. T. To, Jason P. Y. Cheung.

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
