## [Decision Letter · Decision Letter 0]

10 Jun 2020

PONE-D-19-35759

Dose Curve Pattern Impact on the Effects of Physiotherapeutic Scoliosis Specific Exercises on Cobb angle of Adolescent Idiopathic Scoliosis Subjects: A prospective clinical controlled trial with two years follow-up

PLOS ONE

Dear Dr. Fan,

Thank you for submitting your manuscript to PLOS ONE. After careful consideration, we feel that it has merit but does not fully meet PLOS ONE’s publication criteria as it currently stands. Therefore, we invite you to submit a revised version of the manuscript that addresses the points raised during the review process.

We look forward to receiving your revised manuscript.

Kind regards,

Natasha McDonald

Associate Editor

PLOS ONE

Journal Requirements:

2. Thank you for submitting your clinical trial to PLOS ONE and for providing the name of the registry and the registration number. The information in the registry entry suggests that your trial was registered after patient recruitment began. PLOS ONE strongly encourages authors to register all trials before recruiting the first participant in a study.

Please give confirmation that all related trials are registered by stating: “The authors confirm that all ongoing and related trials for this drug/intervention are registered”.

3. You indicated that you had ethical approval for your study.

In your Methods section, please ensure you have also stated whether you obtained consent from parents or guardians of the minors included in the study or whether the research ethics committee or IRB specifically waived the need for their consent.

5. Please amend either the title on the online submission form (via Edit Submission) or the title in the manuscript so that they are identical.

6. Please amend your list of authors on the manuscript to ensure that each author is linked to an affiliation. Authors’ affiliations should reflect the institution where the work was done (if authors moved subsequently, you can also list the new affiliation stating “current affiliation:….” as necessary).

7. Please include your table as part of your main manuscript and remove the individual file.

Please note that supplementary tables should be uploaded as separate "supporting information" files.

8. We note that Figures 3 and 4 include images of patients. 

As per the PLOS ONE policy (http://journals.plos.org/plosone/s/submission-guidelines#loc-human-subjects-research) on papers that include identifying, or potentially identifying, information, the individual(s) or parent(s)/guardian(s) must be informed of the terms of the PLOS open-access (CC-BY) license and provide specific permission for publication of these details under the terms of this license. Please download the Consent Form for Publication in a PLOS Journal (http://journals.plos.org/plosone/s/file?id=8ce6/plos-consent-form-english.pdf). The signed consent form should not be submitted with the manuscript, but should be securely filed in the individual's case notes.

Please amend the methods section and ethics statement of the manuscript to explicitly state that the patient/participant has provided consent for publication: “The individual in this manuscript has given written informed consent (as outlined in PLOS consent form) to publish these case details”.

9. Please include captions for your Supporting Information files at the end of your manuscript, and update any in-text citations to match accordingly. Please see our Supporting Information guidelines for more information: http://journals.plos.org/plosone/s/supporting-information

Reviewers' comments:

Reviewer's Responses to Questions

**Comments to the Author**

1. Is the manuscript technically sound, and do the data support the conclusions?

Reviewer #1: Partly

Reviewer #2: Partly

Reviewer #3: Partly

2. Has the statistical analysis been performed appropriately and rigorously? 

Reviewer #1: Yes

Reviewer #2: No

Reviewer #3: No

3. Have the authors made all data underlying the findings in their manuscript fully available?

Reviewer #1: Yes

Reviewer #2: Yes

Reviewer #3: Yes

4. Is the manuscript presented in an intelligible fashion and written in standard English?

Reviewer #1: Yes

Reviewer #2: Yes

Reviewer #3: Yes

5. Review Comments to the Author

Reviewer #1: The article entitled “Dose Curve Pattern Impact on the Effects of Physiotherapeutic Scoliosis Specific Exercises on Cobb angle of Adolescent Idiopathic Scoliosis Subjects: A prospective clinical controlled trial with two years follow-up” has a good topic and matter in the field. However, the methodology of the study needs to be revised. The authors investigate the effects of scoliosis-specific exercise method on different curve patterns (major thoracic and major lumbar) and in long term. The idea is valuable. But the cobb angle range is wide, therefore the inevitable effect of brace wearing for some participants on the results looks unnoticed. Randomization is needed for age, gender, cobb angle, curve pattern, brace wearing, in-brace correction, risk of progression... etc. Before consideration for publication, there may be following suggestions to consider again:

Abstract

- In method, information regarding Cobb angle of the subjects are needed. Was curve magnitude similar between thoracic and lumbar curve groups?

- Did patient undergo only exercise intervention or did they use bracing as well?

- The first sentence of the background needs to be revised related with exercise. The statement is associated with brace and surgery.

- What were patients’ ages? Was age similar between groups?

Introduction

- The importance of curve progression risk in conservative treatment was extensively reported. But did you randomize your patients in terms of bone maturity among groups?

- Did you measure curve flexibility? If so, how did you measure? Beucause you reported that this study aimed to investigate the long-term therapeutic effect of PSSE on thoracic and lumbar major curves and to determine whether a more flexible lumbar curve would respond better to exercise.

Methods

- One of your inclusion criteria was having cobb angles: from 10 degrees to 50 degrees. This means that some of your patients had bracing indication. How many of your patients did wear brace in this period? Then how did you differentiate exercise specific effect?

- Please add detailed explanation about The Rigo Scoliosis Classification system.

- You included major thoracic and major lumbar curves. I see from the method that some of them, who had major thoracic curver, had double curves or secondary lumbar curves. This would affect results. We know that single lumbar curves and double curves are more common than other curve pattern in idiopathic scoliosis. Single thoracic curves are rare. Therefore, curve distribution among groups would differ. In table 1, curve pattern was reported as double and single, only. But more information is needed regarding curve pattern distribution. For single curve, how many patient had thoracic, how many patient had lumbar? For double curve, how many had primary thoracic? how many had primary lumbar? What was curve pattern distribution of the participants?

- How was curve range for thoracic and lumbar curves? How many vertebrae did curves include? This would affect results. What do you think?

- Did you measure axial trunk rotation of the participants? Curve magnitude would affect from trunk rotation parameter as much as Cobb angle?

- Did you use and patient-reported outcome measures? or Did you assess any clinical parameters such as trunk symmetry and cosmetic deformity?

- Did any of your patient use spinal brace in this period? When considering wide range of Cobb angle, this looks inevitable. How was the distribution of spinal brace usage among groups? How was in-brace correction for patients who underwent spinal brace intervention? Because if the correction amount differ among groups, this would increase or decrease the exercise effect.

- Male distribution is higher in group B. Do you think this would affect results? Also bone maturity distribution among groups seems to be different with 11 - 11 in group A, and 12-6 in group B. Group B looks to have participant who had low risk of progression.

- In table one, minimum and maximum values are needed.

- Did authors use any randomization method for age, gender, Cobb angle, curve progression risk (bone maturation)? If not, how would it be possible to make exact comparison of two different curve pattern groups (thoracic and lumbar)

- I am sendin my suggestions until here. Discussion is also needed to be revised according to methodology.

Reviewer #2: PONE-D-19-35759

The curve title should be corrected to:

Does Curve Pattern Impact the Effects of Physiotherapeutic Scoliosis Specific Exercises on Cobb angles Participants with Adolescent Idiopathic Scoliosis: A prospective clinical controlled trial with two years follow-up.

It is best to refer to participants or persons with a diagnosis rather than subjects. Throughout the text please replace patients by participants or persons.

ABSTRACT

Please specify details of the intensive frequency and reduce frequency. (number of visits per week or month and their duration of exercising at minimum).Specify if a home program was performed.

If the study is controlled. I assumed there are two groups. Specify what the control group did during the trial in the abstract.

Then the analysis should likely be a mixed model ANOVA with a between subject group factor and a within subject (repeated measures) time factor. There should not be multiple paired t-tests if you choose a relevant pairwise comparison over time within curve patterns. Using separate t-tests would increase your risk of type one error.

There are numerous English issues. I will not point all of them as English is a second language for me too. Please have the paper reviewed by an English editor.

EG always use major thoracic curve and major lumbar curve.

Specify the recruitment setting and method for the participants.

Beyond stating no difference in the curve magnitude between groups please report the curve means and SD and how many patients if any were braced. If none wore a brace, specify.

Spell out d on first use for D-value.

Since we don’t know how you calculated d, we don’t know in the abstract if the change over 1 year is a improvement or deterioration in Cobb angles in each group. The sentence with the change in COBB for the specific intervals should specify which group had which results. There should be multiple F statistics and p-values with your design: one for the interaction between groups and time, one for the main effect of group and one for the main effect of time.)

The value for the regressions in group B do not seem to map to any of the values in the sentence prior. How were these group be reductions calculated.

Report data on the compliance and drop outs with the exercise program. Did all 40 participants attend all follow-up. If yes specify this is a per protocol analysis and report how many had started and were not included in the analyses.

The conclusion is inaccurate: This was the first study to investigate whether PSSE can lead to curve

regression based on location of the major curve. The following study examined the effect of curve type in a multivariate analysis of the short-term effects of Schroth on Cobb angles.

Schroth Physiotherapeutic Scoliosis-Specific Exercises Added to the Standard of Care Lead to Better Cobb Angle Outcomes in Adolescents with Idiopathic Scoliosis - an Assessor and Statistician Blinded Randomized Controlled Trial.

Schreiber S, Parent EC, Khodayari Moez E, Hedden DM, Hill DL, Moreau M, Lou E, Watkins EM, Southon SC.

PLoS One. 2016 Dec 29;11(12):e0168746. doi: 10.1371/journal.pone.0168746. eCollection 2016.

Introduction

In the intro where you state the following about the reviews it may be important to review each paper in those reviews for discussions of the effects of curve type. The reviews did not address this question but some of the original reviewed trials did. however, in these studies, the relationship of curve location with correction effects was not clearly discussed, and there was only a short-term followup.

There may be limitations to those as short follow-ups which would still justify your work that you could highlight.

The following intro statement is also overreaching. Schreiber et al above did include baseline Cobb angle in their study of the effect of Schroth. There maybe limitations to this understanding but it is not completely unknown. You stated “the influence of curve magnitude on exercise outcomes is unknown.”

The objective as stated focuses on curve magnitude and flexibility rather than relation between outcomes and curve types. Your focus seems to be on curve type in the abstract. Please re-center the objective on this topic.

Specify your recruitment methods and setting. Were all consecutive eligible participants invited? Only referrals from some DRs…

Please add a justification to the exclusion of hypermobility. This has not been recommended by SOSORT and has not been done in other study. Maybe, if possible, report how many patients were excluded on this basis.

Please specify if patients could have been prescribed a brace at baseline with the exercises or not in the selection criteria.

From a Asklepios (German) Schroth trained perspective grouping N3N4 with 3C is not consistent with only grouping thoracic curves. Please acknowledge as a limitation that the N3N4 group likely had both thoracic and lumbar curves and may have needed different treatment than the 3C curves. For example the Chest twister exercise is not indicated for N3N4. Similarly. The type 4C curves do have a lumbar dominance but also have a thoracic curves. What did you do with the pure lumbar or thoracolumbar curves? Specify if you had any in your sample.

Please specify the certification of the Schroth therapists BSPTS -rigo, or Asklepios or ISST or Weiss…

When were participants asked to complete the simple home exercise compliance questionnaire (Weekly, monthly, once at then of the study…)

In listing exercises could the figure order be arranged to match the order of presentation in the text.

There is a variation of the muscle cylinder for thoracic curves. Specify why it was not used.

Describe how the those was chosen. Did the patient do the same exercise throughout the program? Acknowledge as a limitation why you did not use dynamic exercises such as walking or …

Throughout the paper : I recommend avoiding the labels A and B for the groups and clearly stating thoracic vs Lumbar groups.

Figure 5 and 6 Please add a justification for why these participants were not braced in addition to doing the exercises.

Table 1. Add units for age.

Replace gender by Sex. I doubt you documented gender.

Why not report count and percentage as you did for sex instead for Sanders stage.

Can you specify brace types. This is a notable imbalance and surprisingly more braced in thoracic group?

For future meta-analysis purpose it may be valuable to still measure thoracic and lumbar curves in both groups.

Rather than or in addition to, please report the number of N3N4 vs 4C per group or add a note to each column showing the thoracic double curves were all N3N4 and the Lumbar were all 4C.

Specify in the methods if the classification was clinical only or informed by radiographs also.

Can you report if all thoracic curves were right side and all lumbar were left sides. If not please report the distribution.

Figure 2 could present statistical significance of the comparison of the values as well with symbols.

Specify what the error bar represent (standard deviations or standard errors).

Figure 3. It is unfortunate that the patient has a wide racer back bra which hides much of the postural defect in thoracic spine. The semi-hanging picture has allowed too much hypokyphosis. Would you have a semi-hanging example where the sagittal kyphosis is better maintained. The aspect ratio of the photo f seems off (too wide not tall enough).

Can you add the left right marker to the radiograph. It show right thoracic to the right of the image.

Figure 4. Can you add the left right marker to the radiograph. This one shows the right thoracic curve on the left of the image. Once again in C the patient shows hypokyphosis and this time has lost physiological lordosis. Can you find a demonstration of semi-hanging in this group with better sagittal profile. For G, can you have an example with proper head alignment.

Figure 5 and 6 were presented with figure 6 first. No need to use initials in the figures. Label as example of a thoracic vs lumbar participants. Could you draw the Cobb measurements on the images? Could the scaling of the two spine images in fig 6 be made more similar?

The dataset uploaded should also include a data dictionary or a legend. What values are coded 1 and 0 … Define cut point 30 deg. Is the data incomplete. What appears in columns V to AO??

Specify in the selection criteria what was the criteria for brace prescription and brace termination and how you monitored brace compliance. Detail the kinds of brace that were prepared and possibly the targeted or achieved in brace correction if available.

Can you detail the positioning instructions for the participants during the radiographs.

In clnical trials the study is not powered to compare group characteristics at baseline. It does not mean much to do statistical comparisons of those. Key is whether the estimate appear to present clinical differences before deciding whether to control for those.

Clarify how the d-values were computed. Why not describe it as a mixed model anova with a between group factor and a time factor and use appropriate pairwise characteristics. It appears to be what you did given the output provided. Can you specify which pairwise was requested? LSD?

The use of multiple t-tests to detect regression increases the type one error chances.

Results:

Subjects. I disagree that groups were sufficiently similar for bracing.

From the selection criteria I believed no participant could have started any scoliosis treatment before enrollment. Here I learn that bracing could have been started prior to PSSE. This should be clear from the patient selection criteria.

Can you report brace compliance in both groups?

Why only report the percent radiograph refused at 6mth into PSSE. Why not also for the other time points.

Specify if the following statement applies only to group averages or also to patient individual data: There was no deterioration of the major curvature in either group at the 2-year follow-up.

Where you report the F and P-values you need three F values. The key one is for the interaction between group and time, then you also need main effects on groups and on time. (your output show all were not significant.

Report the Sanders secondary analysis as a new paragraph. Announce this interest in objective section as a secondary objective.

If the interaction effect is not significant it is not good statistical practice to explore pairwise within groups with t-tests. IF you had set your mixed method anova with Cobb at baseline, 1 year, 1.5 yrs and 2 yrs you may have detected this effect already. Running a different approach with t-test exposes you to a higher risk of type 1 errors.

It would be possible to use a chi-square analysis to compare the distribution of the improved, stabilized and deteriorated in each group statistically in addition to just report the results. A statistician may be able to determine if the three time points could be compared in a single analysis for these proportions to protect against a type one error.

See above. The following opening statement of the discussion is overreaching: This is the first study to investigate the outcomes of different curve types undergoing PSSE treatment for AIS.

You cannot state the following as all your groups received exercises. You did not show superiority to an alternative. And you did not discuss historical controls. Therefore, our results further revealed the superiority of PSSE programs in the stabilization and regression of scoliosis.

In the discussion you state: This met our original study hypothesis that lumbar major curves are more flexible. I am not sure this is the only possible physiological explanation. You did not measure flexibility per say. Please justify your statement and address possible alternatives as well. Discuss limitations of not having measured flexibility directly.

P13. Compare your results of lumbar effects being better to those of Schreiber et al. I believe they did not find the same curve pattern as the one with the best response. Could it be that some benefits for your thoracic group may have been missed due to not using the chest twister and prone/supine exercises…

Please discuss the limitation of self-reporting compliance and possible recall issues depending on when your participants completed the compliance questionnaires.

Limitations. There are more than 4 patterns in RIGO classification. Type 1 and type 2 were not discussed here.

Please discuss not monitoring co-interventions such as manipulations, massage, other fitness activities or self-used of off the shelve bracing.

Could you review how many patients in each group experience a change of curve type over time and report which changes occurred.

Discuss the imbalanced in the number of braced participant and possible brace compliance effects on results.

Discuss whether after 2 years of follow-up these participants had reached skeletal maturity. How many had reached discharge point or were sufficiently far after peak growth velocity to deem the results final or not.

The SRS-SOSORT recommendations propose a number of analysis reporting guidelines. Could you report Risser signs in your dataset or table 1. Could you report number of patients with curve over 30 at each time point. I believe none exceeded surgery threshold of 45 or 50 but this could be specified.

The discussion could compare how your follow-up length and results compared to other Schroth studies. Are your results better than others or similar. Was the progression risk of your cohort similar or worse than others.

Please add a paragraph about whether there is a risk of overtreatment in this cohort. Could some of the patients have been left alone and avoid the burden of treatment. I can see that some would argue that for the fact that these patients possibly had no risk of progressing to bracing or surgery. However you showed some notable regression. This may appeal to some participants with small curves at low risk of progression. Still the possible overtreatment issue should be discussed.

An important limitation in your study is that you may have been underpowered to detect difference between groups. It would be important to discuss how big a different in effects between curve types may be clinically important. This may be difficult to determine but a discussion of this topic should be presented.

I would recommend against including the following in the discussion as you did not measure flexibility and did not study relation of effects of PSSE with baseline curve magnitude per say. However, further studies are necessary to address the correlation between spinal flexibility and the correction effects of PSSE at different curve magnitudes. IF you wish to keep then please move earlier in the discussion as suggestions for future research.

I have annotated a few elements of the paper in an uploaded copy.

Reviewer #3: The objective of this study is to investigate the comparative effectiveness of the PSSE correction effect on the Cobb angle between the thoracic and lumbar curves in AIS subjects. The authors considered a prospective clinical controlled study. While the study objectives sound interesting, a number of shotcomings were observed, in regards to abiding by the CONSORT guidelines for conducting and reporting results of high-quality randomized controlled trials (RCTs).

1. Abstract:

The authors state results as: "A significant Cobb angle reduction was observed...", without any statement of p-values, estimated effect size, and its precision as confidence intervals, or CIs (say, 95\\%). This can appear confusing, and half-baked to a reader. Check CONSORT checklist for Abstracts reporting of RCTs, and rewrite the Abstract following guidelines.

2. Methods:

Methods reporting appeared very messy. An orderly manner is suggested, following CONSORT guidelines, without repeating information, such as Trial Design, Participant Eligibility Crtieria and settings, Interventions, Outcomes, sample size/power considerations, Interim analysis and stopping rules, Randomization (details on random number generation, allocation concealment, implementation), Blinding issues, etc. The authors are advised to create separate subsections for each of the possible topics (whichever necessary), and that way produce a very clear writeup.

(a) For instance, the randomization and allocation concealment should be made very clear; the trial staff recruiting patients should not have the randomization list. Randomization should be prepared by the trial statistician, and he/she would not participate in the recruiting. I am confused; was randomization not done during the "Allocation" phase in your CONSORT diagram?

(b) I am surprized to see no statement on sample size/power in a manuscript proposing a (clinical) controlled trial. This is really the key here!

(c) t-tests were used for assessing group differences for continuous variables (under the assumption of Normality). What if underlying normality assumptions are violated? Why not non-parametric (robust) tests were proposed?

(d) Similarly, the one-way repeated measures ANOVA may also be replaced/presented by a nonparametric Friedman-type test. I mean, justification is needed for the underlying Normality assumptions.

3. Results & Conclusions:

(a) The authors should check that any statement of significance should be followed by a p-value in the entire Results section.

(b) The authors admitted a long list of limitations in their work, notably, the "...unable to blind the subjects, or physiotherapists...". Despite the justification provided, I am not sure how good the trial is! With 40 subjects recruited, the results stated, at best, can only be claimed as from a pilot study. This needs to be clearly stated, and the study cannot be claimed as a nicely planned randomized trial. Also, they need to state that future studies (with larger sample sizes) are warranted to really understand the comparative efficiacy.

6. PLOS authors have the option to publish the peer review history of their article (what does this mean?). If published, this will include your full peer review and any attached files.

Reviewer #1: No

Reviewer #2: Yes: Eric Parent

Reviewer #3: No

---

## [Author Response · Author response to Decision Letter 0]

4 Aug 2020

REPLY TO REVIEWERS

Re: Ms. No. PONE-D-19-35759 - "Dose Curve Pattern Impact on the Effects of Physiotherapeutic Scoliosis Specific Exercises on Cobb angle of Adolescent Idiopathic Scoliosis: A prospective clinical controlled trial with two years follow-up”

Dear Editor, 

The authors would like to thank you and the Reviewers for all of your time and effort devoted to the review of our aforementioned manuscript. The Reviewers’ comments were indeed extremely insightful and greatly appreciated. As such, the authors would like to take this opportunity to address each and every concern the Reviewers noted in their review of our submission. In addition, where appropriate, we have also revised our manuscript accordingly. Major changes are made in light gray.

We believe that the Reviewers’ comments have helped improve the quality of our manuscript. We hope that you and the Reviewers will find our revised work suitable for publication in PLOS One. 

Editor

Comment #1:  Journal Requirements:

Response: We have revised the format accordingly.

Comment #2: 2. Thank you for submitting your clinical trial to PLOS ONE and for providing the name of the registry and the registration number. The information in the registry entry suggests that your trial was registered after patient recruitment began. PLOS ONE strongly encourages authors to register all trials before recruiting the first participant in a study.

Please give confirmation that all related trials are registered by stating: “The authors confirm that all ongoing and related trials for this drug/intervention are registered”.

Response: We have revised this. Please see lines 108-110, page 7 (Materials and methods: study design)

Comment #3: 3. You indicated that you had ethical approval for your study.

In your Methods section, please ensure you have also stated whether you obtained consent from parents or guardians of the minors included in the study or whether the research ethics committee or IRB specifically waived the need for their consent.

Response: Thank you for the comment. Please see lines 105-107,page 7 (Materials and methods: study design)

Comment #4: 4. We note that you have stated that you will provide repository information for your data at acceptance. Should your manuscript be accepted for publication, we will hold it until you provide the relevant accession numbers or DOIs necessary to access your data. If you wish to make changes to your Data Availability statement, please describe these changes in your cover letter and we will update your Data Availability statement to reflect the information you provide.

Response: No change is needed.

Comment #5: 5. Please amend either the title on the online submission form (via Edit Submission) or the title in the manuscript so that they are identical.

Response: Thank you this has been changed. The title is now: Dose curve pattern impact on the effects of physiotherapeutic scoliosis specific exercises on Cobb angles of participants with adolescent idiopathic scoliosis: a prospective clinical trial with two years follow-up

Comment #6: 6. Please amend your list of authors on the manuscript to ensure that each author is linked to an affiliation. Authors’ affiliations should reflect the institution where the work was done (if authors moved subsequently, you can also list the new affiliation stating “current affiliation:….” as necessary).

Response: This has been done.

Comment #7: 7. Please include your table as part of your main manuscript and remove the individual file.

Please note that supplementary tables should be uploaded as separate "supporting information" files.

Response: This has been added.

Comment #8: 8. We note that Figures 3 and 4 include images of patients. 

As per the PLOS ONE policy (http://journals.plos.org/plosone/s/submission-guidelines#loc-human-subjects-research) on papers that include identifying, or potentially identifying, information, the individual(s) or parent(s)/guardian(s) must be informed of the terms of the PLOS open-access (CC-BY) license and provide specific permission for publication of these details under the terms of this license. Please download the Consent Form for Publication in a PLOS Journal (http://journals.plos.org/plosone/s/file?id=8ce6/plos-consent-form-english.pdf). The signed consent form should not be submitted with the manuscript, but should be securely filed in the individual's case notes.

Please amend the methods section and ethics statement of the manuscript to explicitly state that the patient/participant has provided consent for publication: “The individual in this manuscript has given written informed consent (as outlined in PLOS consent form) to publish these case details”.

Response: Thank you for the comment. We have included the necessary documents. The statement was made on lines 107 to 108, page 7 (Materials and methods: study design).

Comment #9: 9. Please include captions for your Supporting Information files at the end of your manuscript, and update any in-text citations to match accordingly. Please see our Supporting Information guidelines for more information: http://journals.plos.org/plosone/s/supporting-information

Response: These have been included.

Reviewer #1

Comment #1: The article entitled “Dose Curve Pattern Impact on the Effects of Physiotherapeutic Scoliosis Specific Exercises on Cobb angle of Adolescent Idiopathic Scoliosis Subjects: A prospective clinical controlled trial with two years follow-up” has a good topic and matter in the field. However, the methodology of the study needs to be revised. The authors investigate the effects of scoliosis-specific exercise method on different curve patterns (major thoracic and major lumbar) and in long term. The idea is valuable. But the cobb angle range is wide, therefore the inevitable effect of brace wearing for some participants on the results looks unnoticed. Randomization is needed for age, gender, cobb angle, curve pattern, brace wearing, in-brace correction, risk of progression... etc. Before consideration for publication, there may be following suggestions to consider again:

Abstract - In method, information regarding Cobb angle of the subjects are needed. Was curve magnitude similar between thoracic and lumbar curve groups?

Response: We thank the reviewer for the comments and we will try to amend our manuscript accordingly. 

The curve magnitude was similar between groups: 28.5 ± 8.4 vs 26.8 ± 10.1, p=0.6. 

Please refer to the 1) page 3, abstract: results: lines 29 - 31; 

2)results, participants, lines 222 -227; 

3)Table 1: line 237

Comment #2:  Did patient undergo only exercise intervention or did they use bracing as well?

Response: A total of 10 participants were wearing brace before commencing the PSSE program and having comparable, 37% of correction effects from bracing. 

Please refer to the 1) results: participants, lines 231 - 235. 2) Table 1: line 237

Comment #3: The first sentence of the background needs to be revised related with exercise. The statement is associated with brace and surgery.

Response: We thank the reviewer for this suggestion. This has been revised. Please refer to page 2, abstract: background, lines 10-11

Comment #4: What were patients’ ages? Was age similar between groups?

Response: Yes, the age was under evenly distributed and was similar between groups.

Please refer to the page 3, abstract: results, lines 30 to 31, Table 1 (line 237) and lines 227 to 228 of page 15.

Comment #5: Introduction

- The importance of curve progression risk in conservative treatment was extensively reported. But did you randomize your patients in terms of bone maturity among groups?

Response: Thanks for comment. This study was planned as a case control study. Participants were recruited from one scoliosis clinic and grouped into two groups according to their curve pattern. The progression related data was analyzed at the baseline and correlation analysis was performed to detect any influence of these factors, including bone maturity, on the reduction of Cobb angles. Please refer to the methodology, study design, lines 119-123, line 125 (S2 Fig. 1: CONSORT flow diagram), line 237 (Table 1) and lines 210 to 213 (statistical analysis).

Comment #6: Did you measure curve flexibility? If so, how did you measure? Because you reported that this study aimed to investigate the long-term therapeutic effect of PSSE on thoracic and lumbar major curves and to determine whether a more flexible lumbar curve would respond better to exercise.

Response: This is an important point by the reviewer. Unfortunately we were unable to quantify spinal flexibility prior to treatment in this study due to patients’ radiation concern, therefore, the hypothesis was re-centered on curve patterns. Please see the text at lines 94 to 99 (Introduction), lines 369 to 374 (discussion); lines 433 to 441 (Study limitation). 

Comment #7: Methods

- One of your inclusion criteria was having cobb angles: from 10 degrees to 50 degrees. This means that some of your patients had bracing indication. How many of your patients did wear brace in this period? Then how did you differentiate exercise specific effect? 

Response: Ten participants (5 were wearing full time brace whereas 5 were wearing only night time brace), were wearing brace and achieved comparable in-brace correction 37%, which explained the effects from bracing and allow this study emphasized on the effect from the PSSE.

Please check the Table 1 (line 237), lines 231 to 235 (results) and lines 349 to 353 (discussion).

Comment #8: Please add detailed explanation about The Rigo Scoliosis Classification system.

Response: Rigo clasification was adopted to rigo cheneau brace fabrication and also BSPT Schroth methods, in which 5 classifications with 8 sub-types were clearly introduced. two important references was cited accordingly, please see references 22 and 23.

Comment #9: You included major thoracic and major lumbar curves. I see from the method that some of them, who had major thoracic curve, had double curves or secondary lumbar curves. This would affect results. We know that single lumbar curves and double curves are more common than other curve pattern in idiopathic scoliosis. Single thoracic curves are rare. Therefore, curve distribution among groups would differ. In table 1, curve pattern was reported as double and single, only. But more information is needed regarding curve pattern distribution. 

Response: We thank the reviewer for these comments. This information was revised in:

1.Lines 222 to 227.

2.S2: Fig.1 CONSORT flow chart

3.Table 1. 

4.Please also see lines 416-426 of paragraph-study limitation.

Comment #10: For single curve, how many patient had thoracic, how many patient had lumbar? For double curve, how many had primary thoracic? how many had primary lumbar? What was curve pattern distribution of the participants?

Response: Thanks for your suggestion. The additional information has been added to:.

1.Lines 222 to 227.

2.S2: Fig.1 CONSORT flow chart

3.Table 1. 

Please also see lines 416-426 of paragraph-study limitation.

Comment #11: How was curve range for thoracic and lumbar curves? How many vertebrae did curves include? This would affect results. What do you think?

Response: Fantastic comment, we do agree that the range of the curvature would definitely impact on the treatment effects, in terms of that may affect the spinal segmental flexibility. However, curve range should also be correlated with curve pattern, such as a short range should be observed in single lumbar curves whereas a long range should be observed in a thoracolumbar, etc. This idea is good but unfortunately out of scope of this study, the sample size of each curve pattern was small in our study, therefore, extra analysis of range of curvature may cause distraction from our primary outcome. Moreover, it will be too informative and overwhelming to readers of those who are not familiar with scoliosis radiographic classification. However, it will be interesting to study the effects of PSSE on the different radiographic parameters. Please see this amendings to lines 446-447 of paragraph conclusion.

Comment #12: Did you measure axial trunk rotation of the participants? Curve magnitude would affect from trunk rotation parameter as much as Cobb angle?

Response: This is an important parameter that will be studied in a future study. We wanted to mainly focus on the Cobb angle changes in this manuscript. Further investigation of the cosmesis and quality of life will also be performed. Please see lines 447-448 of paragraph Conclusion.

Comment #13: Did you use and patient-reported outcome measures? or Did you assess any clinical parameters such as trunk symmetry and cosmetic deformity?

Response: As discussed in comment #12, these are important parameters to study. We have not included them in this manuscript because we wanted to focus only on curve magnitude changes.

Comment #14: Did any of your patient use spinal brace in this period? When considering wide range of Cobb angle, this looks inevitable. How was the distribution of spinal brace usage among groups? How was in-brace correction for patients who underwent spinal brace intervention? Because if the correction amount differ among groups, this would increase or decrease the exercise effect.

Response: Please check the Table 1 and lines 231 to 235; lines 323 to 326 of paragraph results, discussion: lines 349 to 353; lines 405 to 408.

Comment #15: Male distribution is higher in group B. Do you think this would affect results? Also bone maturity distribution among groups seems to be different with 11 - 11 in group A, and 12-6 in group B. Group B looks to have participant who had low risk of progression.

Response: These are important confounders. Using the logistic correlation analysis, we did not find any significant relationship with gender and maturity.

Please refer to lines 323 to 326, Table 1.

Comment #16: In table one, minimum and maximum values are needed.

Response: Range was reported in the abstract; Mean ± SD and 95% CI have been added in the Table 1.

Comment #17: Did authors use any randomization method for age, gender, Cobb angle, curve progression risk (bone maturation)? If not, how would it be possible to make exact comparison of two different curve pattern groups (thoracic and lumbar)

Response: It was convenient sampling with a curve pattern controlled study, the assumption of normality was calculated before commencing the study and analysis was conducted for baseline data between groups. Logistic regression analysis was adopted to detect any correlation of variables that differed on baseline between groups and treatment effect, which was reported on lines 210 to 213; lines 323 to 326, curve distribution can be found in S2 Fig 1 and table 1.

Comment #18: I am sending my suggestions until here. Discussion is also needed to be revised according to methodology.

Response: Thanks for your valuable suggestions and hope you will find the updated discussion interesting. 

Reviewer #2

Comment #1: The curve title should be corrected to: Does Curve Pattern Impact the Effects of Physiotherapeutic Scoliosis Specific Exercises on Cobb angles Participants with Adolescent Idiopathic Scoliosis: A prospective clinical controlled trial with two years follow-up.

Response: We thank the reviewer for this suggestion and this has been revised.

Comment #2: It is best to refer to participants or persons with a diagnosis rather than subjects. Throughout the text please replace patients by participants or persons.

Response: These have been corrected.

Comment #3: ABSTRACT Please specify details of the intensive frequency and reduce frequency. (number of visits per week or month and their duration of exercising at minimum). Specify if a home program was performed.

Response: This has been done. Please refer to the page 2, abstract, methods, lines 19 to 23.

Comment #4: If the study is controlled. I assumed there are two groups. Specify what the control group did during the trial in the abstract.

Response: This was planned as a clinical, curve pattern controlled study. Please refer to the page 2, abstract, methods, lines 17 to 19. S2 Fig 1 Consort flow diagram.

Comment #5: Then the analysis should likely be a mixed model ANOVA with a between subject group factor and a within subject (repeated measures) time factor. There should not be multiple paired t-tests if you choose a relevant pairwise comparison over time within curve patterns. Using separate t-tests would increase your risk of type one error.

Response: We thank the reviewer for this astute comment. The analyses have been corrected according to your valuable suggestions, please see pages 2-3: abstract, method (abstract): lines 24 to 26; statistics lines (main text): 196 to 213; results (abstracts): line 34-39; results (main text): lines 259 to 291; Tables 2,3. 

Comment #6: There are numerous English issues. I will not point all of them as English is a second language for me too. Please have the paper reviewed by an English editor.

EG always use major thoracic curve and major lumbar curve.

Response: Thank you for the comments. This has been amended accordingly.

Comment #7: Specify the recruitment setting and method for the participants.

Response: This has been done. Please see page 7, methods, study design, lines 103 to 105; lines 110 to 123 and S2 (Fig. 1: CONSORT flow diagram).

Comment #8: Beyond stating no difference in the curve magnitude between groups please report the curve means and SD and how many patients if any were braced. If none wore a brace, specify.

Response: Ten participants (5 were wearing full time brace whereas 5 were wearing only night time brace), were wearing brace and achieved comparable in-brace correction 37%, which explained the effects from bracing and allow this study emphasized on the effect from the PSSE.

Please check lines 132 to 135 of page 9, the table 1 and results, lines 231 to 235; discussion, lines 348 to 351.

.

Comment #9: Spell out d on first use for D-value.

Response: The D-value stands for the difference value of Cobb angle over three testing times with the initial Cobb angle, to detect the regression, stabilization and deterioration between groups. This has been reiterated in lines 203-206 and table 2.

Comment #10: Since we don’t know how you calculated d, we don’t know in the abstract if the change over 1 year is a improvement or deterioration in Cobb angles in each group. The sentence with the change in COBB for the specific intervals should specify which group had which results. There should be multiple F statistics and p-values with your design: one for the interaction between groups and time, one for the main effect of group and one for the main effect of time.)

Response: We are sorry for this confusion and the analysis has been clarified. Please see the revised abstract, methods (lines 203-206), results (lines 259 to 365) and Table 2.

Comment #11: The value for the regressions in group B do not seem to map to any of the values in the sentence prior. How were these group be reductions calculated.

Response: Regression was defined as a decrease of Cobb angle ≥ 6 degrees, the percentage value in each group was reported in the results session, in addition, raw Cobb angle was also analyzed using the repeated measures, please refer to the outcome measurement (lines 184 to 186) and results (lines 259-265; lines 272 - 291). All results, including mean, SD, 95% CI of the D-value and of the raw Cobb angles, were reiterated and summarized in the Table 2 and Table 3, respectively. 

Comment #12: Report data on the compliance and drop outs with the exercise program. Did all 40 participants attend all follow-up. If yes specify this is a per protocol analysis and report how many had started and were not included in the analyses.

Response: There were no dropouts through the whole studies. Per-protocol analysis was specified accordingly. Please see lines 253 - 257; S2: Fig.1 CONSORT flow diagram

Comment #13: The conclusion is inaccurate: This was the first study to investigate whether PSSE can lead to curve regression based on location of the major curve. The following study examined the effect of curve type in a multivariate analysis of the short-term effects of Schroth on Cobb angles. Schroth Physiotherapeutic Scoliosis-Specific Exercises Added to the Standard of Care Lead to Better Cobb Angle Outcomes in Adolescents with Idiopathic Scoliosis - an Assessor and Statistician Blinded Randomized Controlled Trial. Schreiber S, Parent EC, Khodayari Moez E, Hedden DM, Hill DL, Moreau M, Lou E, Watkins EM, Southon SC.

PLoS One. 2016 Dec 29;11(12):e0168746. doi: 10.1371/journal.pone.0168746. eCollection 2016.

Response: Thanks for you pointing this out, the conclusion and discussion has been revised to “an extension of previous studies”.

Comment #14: Introduction

In the intro where you state the following about the reviews it may be important to review each paper in those reviews for discussions of the effects of curve type. The reviews did not address this question but some of the original reviewed trials did. however, in these studies, the relationship of curve location with correction effects was not clearly discussed, and there was only a short-term followup. There may be limitations to those as short follow-ups which would still justify your work that you could highlight.

Response: Thanks for this brilliant suggestion, the introduction was revised and followed your suggestions, please refer to the lines 77 to 85 (Introduction).

Comment #15: The following intro statement is also overreaching. Schreiber et al above did include baseline Cobb angle in their study of the effect of Schroth. There maybe limitations to this understanding but it is not completely unknown. You stated “the influence of curve magnitude on exercise outcomes is unknown.”

Response: We apologize for the oversight. This has been corrected.

Comment #16: The objective as stated focuses on curve magnitude and flexibility rather than relation between outcomes and curve types. Your focus seems to be on curve type in the abstract. Please re-center the objective on this topic.

Response: We thank the reviewer for this comment. It has been re-centered. Please see lines 94 - 98.

Comment #17: Specify your recruitment methods and setting. Were all consecutive eligible participants invited? Only referrals from some DRs…

Response: patients were recruited from a scoliosis clinic, therefore, all consecutive eligible participants were invited and decision was made according to the inclusion/exclusion criteria. Please see lines 111 to 123 and S2 Fig 1.

Comment #18: Please add a justification to the exclusion of hypermobility. This has not been recommended by SOSORT and has not been done in other study. Maybe, if possible, report how many patients were excluded on this basis.

Response: There were 18 subjects were detected with hypermobility, in form of Beighton score >4, who were excluded from this study. In addition, a total of sixty-two subjects were excluded after eligibility assessment, the exclusion reasons were specified on the lines 115-119; lines 218 to 222 (Results: participants) and S2 (Fig.1 CONSORT flow diagram). One previous study reported hypermobility as a dominant presence in the single curve type and should be taken into account during physiotherapy. Participants with hypermobility were excluded to control/limit the systematic error in this study, the related paper was cited in accordingly. Please see reference 21.

Comment #19: Please specify if patients could have been prescribed a brace at baseline with the exercises or not in the selection criteria.

Response: Yes, they could have. Please see line 115 (exclusion criteria: previous treatment) and lines 132-135 (paragraph: study interventions).

Comment #20: From a Asklepios (German) Schroth trained perspective grouping N3N4 with 3C is not consistent with only grouping thoracic curves. Please acknowledge as a limitation that the N3N4 group likely had both thoracic and lumbar curves and may have needed different treatment than the 3C curves. For example the Chest twister exercise is not indicated for N3N4. Similarly. The type 4C curves do have a lumbar dominance but also have a thoracic curves. What did you do with the pure lumbar or thoracolumbar curves? Specify if you had any in your sample.

Response: N3N4, BSPT approach, is a bit different from the German approach, it mainly focuses on correction for thoracic curve, like gentle chest twisting was applicable but should perform without changing the balance and without over correction. By contrast, the double curve in the German approach, takes the lumbar curvature in priority, therefore using pelvic strategy more than chest twisting, etc. The principles are mostly same but the practical metric is a bit different. We did not explain too much or try to compare the two methods in this manuscript, we were trying to prove the effects of Schroth with Cobb correction instead of centering on how did we position patients. Explaining this practical metric is valuable but may be overwhelming for readers who are not familiar with Schroth, moreover, the dynamic feedback of the spine with exercise is unclear yet. Therefore, after all consideration, we did not include this explanation in our manuscript. 

Comment #21: Please specify the certification of the Schroth therapists BSPTS -rigo, or Asklepios or ISST or Weiss…

Response: BSPTS approach, we firstly registered as BSPTS in 2016, then registered Asklepios in 2018, therefore, mainly BSPTS concept in this study. Please see lines 129.

Comment #22: When were participants asked to complete the simple home exercise compliance questionnaire (Weekly, monthly, once at then of the study…)

Response: Monthly. Please see lines 137-143.

Comment #23: In listing exercises could the figure order be arranged to match the order of presentation in the text.

Response: Sorry for the confusion. This has been amended.

Comment #24: There is a variation of the muscle cylinder for thoracic curves. Specify why it was not used.

Response: Please see lines 153-161. This was not considered due to the pain at the knee joint.

Comment #25: Describe how the those was chosen. Did the patient do the same exercise throughout the program? Acknowledge as a limitation why you did not use dynamic exercises such as walking or …

Response: We thank the reviewer for this astute comment. Please see lines 158-161.

Comment #26: Throughout the paper: I recommend avoiding the labels A and B for the groups and clearly stating thoracic vs Lumbar groups.

Response: We thank the reviewer for this comment. It has been changed.

Comment #27: Figure 5 and 6 Please add a justification for why these participants were not braced in addition to doing the exercises.

Response: Two patients were mostly considering about the cosmetic issue; especially one girl was at late Risser stage (Fig 5) and bracing was not compulsory for her, then this girl tried exercise for a half year and resulted with a significant correction, then she had never considered bracing again.

Comment #28: Table 1. Add units for age.

Response: Added. Table 1

Comment #29: Replace gender by Sex. I doubt you documented gender.

Response: This has been done.

Comment #30: Why not report count and percentage as you did for sex instead for Sanders stage.

Response: The Sanders stage has been replaced by Risser, because of the progression risk value in terms of the Lonstein and Carlson score, that needs the Risser for calculation. Please see the S4 File: raw data set and Table 1. 

Comment #31: Can you specify brace types. This is a notable imbalance and surprisingly more braced in thoracic group?

Response: Thanks for the comment. Please see lines 231 - 233 on page 9. All (n=10) were wearing Cheneau brace (Table 1). 

Comment #32: For future meta-analysis purpose it may be valuable to still measure thoracic and lumbar curves in both groups.

Response: We agree with the suggestion. Please see Table 1, S2 (Fig. 1: CONSORT flow Chart) and S4 File: study data. 

Comment #33: Rather than or in addition to, please report the number of N3N4 vs 4C per group or add a note to each column showing the thoracic double curves were all N3N4 and the Lumbar were all 4C.

Response: This has been added. Please see S2 (Fig. 1: CONSORT flow Chart) and the Table 1. 

Comment #34: Specify in the methods if the classification was clinical only or informed by radiographs also.

Response: Radiographic, BSPTS is using this classification system for bracing too, therefore the X-ray is in priority for classification and the clinical presentation was only for clinical reference.

Please see lines 119 to 120 and S2 (Fig.1 CONSORT flow diagram.) 

Comment #35: Can you report if all thoracic curves were right side and all lumbar were left sides. If not please report the distribution.

Response: This has been clarified in lines 222-227, (S2 Fig.1 Consort flow diagram) and Table 1.

Comment #36: Figure 2 could present statistical significance of the comparison of the values as well with symbols.

Specify what the error bar represent (standard deviations or standard errors).

Response: The new figure 1 has been amended. The previous Figure 2 was replaced by the table 3 shows raw data of Cobb angles over four testing time points, in which mean +/- SD, 95% CI and range were all presented, which may be of value for future meta-analysis. 

Comment #37: Figure 3. It is unfortunate that the patient has a wide racer back bra which hides much of the postural defect in thoracic spine. The semi-hanging picture has allowed too much hypokyphosis. Would you have a semi-hanging example where the sagittal kyphosis is better maintained. The aspect ratio of the photo f seems off (too wide not tall enough).

Can you add the left right marker to the radiograph. It show right thoracic to the right of the image.

Response: A new picture has been uploaded. Unfortunately, we cannot find a better photo as patients cannot come to our center due to COVID. Therefore, I used her another pic for demonstration, please check see whether it is acceptable.

Comment #38: Figure 4. Can you add the left right marker to the radiograph. This one shows the right thoracic curve on the left of the image. Once again in C the patient shows hypokyphosis and this time has lost physiological lordosis. Can you find a demonstration of semi-hanging in this group with better sagittal profile. For G, can you have an example with proper head alignment.

Response: This has been amended.

Comment #39: Figure 5 and 6 were presented with figure 6 first. No need to use initials in the figures. Label as example of a thoracic vs lumbar participants. Could you draw the Cobb measurements on the images? Could the scaling of the two spine images in fig 6 be made more similar?

Response: This has been added.

Comment #40: The dataset uploaded should also include a data dictionary or a legend. What values are coded 1 and 0 … Define cut point 30 deg. Is the data incomplete. What appears in columns V to AO??

Response: Done, the file has been amended. The other columns were including other data such as ATR, SRS-22 etc. We have deleted them as the data on radiographic measurements were used in this study. Please see S4 File: study data.

Comment #41: Specify in the selection criteria what was the criteria for brace prescription and brace termination and how you monitored brace compliance. Detail the kinds of brace that were prepared and possibly the targeted or achieved in brace correction if available.

Response: The SRS bracing criteria was used by doctors to prescribe brace treatment, see lines 132 to 135, and the reference was cited accordingly. Brace termination was defined with skeletal maturity, and bracing compliance was monitored using a monthly self-reported adherence checklist, see lines 139 to 143. In-brace correction and brace type were clarified on lines 233 to 235 and Table 1. 

Comment #42: Can you detail the positioning instructions for the participants during the radiographs.

In clinical trials the study is not powered to compare group characteristics at baseline. It does not mean much to do statistical comparisons of those. Key is whether the estimate appear to present clinical differences before deciding whether to control for those.

Response: amended accordingly. Please see line 144 to 153. Picture legend of Fig. 2 and Fig. 3.

The comparison of the baseline data was to show the assumption of normality, if the baseline was evenly distributed and which can power our study quality. However, I fully agree with you that a clinical trial should look more on the factors that may affect results, but we have to do so according to suggestions from other reviews. In this study, with no deterioration after 2 years, the logistic regression was conducted to analyze the relationship of age, sex, brace etc. and treatment result, which further explained the influence of aforementioned factors on the Cobb reduction. See lines 323 to 326 (Results).

Comment #43: Clarify how the d-values were computed. Why not describe it as a mixed model anova with a between group factor and a time factor and use appropriate pairwise characteristics. It appears to be what you did given the output provided. Can you specify which pairwise was requested? LSD? The use of multiple t-tests to detect regression increases the type one error chances.

Response: LSD was used. We have corrected the analyses appropriately. We thank the reviewer for these valuable suggestions.

Comment #44: Results:

Subjects. I disagree that groups were sufficiently similar for bracing.

Response: Our comment was based on the results of the X2 test which showed no statistical difference. Nevertheless, logistic regression was performed to test whether bracing was an impact on the results. See lines 323 to 326 (Results).

Comment #45: From the selection criteria I believed no participant could have started any scoliosis treatment before enrollment. Here I learn that bracing could have been started prior to PSSE. This should be clear from the patient selection criteria.

Response: We thank for reviewer for this valuable suggestion. The patients started with either bracing or exercise would be excluded in this study, please see S2 (Fig. 1) and lines 132 - 135.

Comment #46: Can you report brace compliance in both groups?

Response: Please see line 233 to 235 and Table 1, because of there was no correlation of bracing with regression effects, the bracing hour, bracing compliance from those 10 participants were not further discussed. 

Comment #47: Why only report the percent radiograph refused at 6mth into PSSE. Why not also for the other time points.

Response: All participants completed radiographic assessment on the baseline, 1-year, 1.5-year and 2-year follow-up, therefore per-protocol analysis was performed. Please see lines 256 – 257. lines 272 to 273 and S2 Fig.1.

Comment #48: Specify if the following statement applies only to group averages or also to patient individual data: There was no deterioration of the major curvature in either group at the 2-year follow-up. 

Response: It applies to both average and individual data. No deterioration was noted for either group. Please see lines 259 to 260 and Table 2.

Comment #49: Where you report the F and P-values you need three F values. The key one is for the interaction between group and time, then you also need main effects on groups and on time. (your output show all were not significant.

Response: The analysis was re-performed by a statistician who was blinded to this study. We thank the reviewer for this important comment. Three F values of comparisons in D-value of Cobb and raw data of Cobb, respectively, were clarified and reported. See lines 263 to 265, table 2,3,4 and lines 275 – 277.

Comment #50: Report the Sanders secondary analysis as a new paragraph. Announce this interest in objective section as a secondary objective.

Response: The Sanders staging has been replaced by Risser as we needed the Risser for evaluation of progression risk.

Comment #51: If the interaction effect is not significant it is not good statistical practice to explore pairwise within groups with t-tests. IF you had set your mixed method anova with Cobb at baseline, 1 year, 1.5 yrs and 2 yrs you may have detected this effect already. Running a different approach with t-test exposes you to a higher risk of type 1 errors.

Response: Fully agree, beside of analysis of D-value of Cobb reduction over three testing times, we re-performed analysis on the Raw Cobb angle over four testing timepoints which suggested a significant time effects in reducing Cobb angles (repeated one way anova).

Comment #52: It would be possible to use a chi-square analysis to compare the distribution of the improved, stabilized and deteriorated in each group statistically in addition to just report the results. A statistician may be able to determine if the three time points could be compared in a single analysis for these proportions to protect against a type one error.

Response: We have proceeded to the reviewer’s suggestions. Unfortunately, the X2 analysis also showed no significance, therefore, we further analyzed the relationship of influencing factors, including age, brace, sex etc and the regression effects using logistic regression analysis. See lines 210 to 213 and lines 323 to 326. 

Comment #53: See above. The following opening statement of the discussion is overreaching: This is the first study to investigate the outcomes of different curve types undergoing PSSE treatment for AIS.

Response: The statement has been corrected. Please see lines 341 -342 (discussion).

Comment #54: You cannot state the following as all your groups received exercises. You did not show superiority to an alternative. And you did not discuss historical controls. Therefore, our results further revealed the superiority of PSSE programs in the stabilization and regression of scoliosis.

Response: We thank the reviewer for the sentiments.

Comment #55: In the discussion you state: This met our original study hypothesis that lumbar major curves are more flexible. I am not sure this is the only possible physiological explanation. You did not measure flexibility per say. Please justify your statement and address possible alternatives as well. Discuss limitations of not having measured flexibility directly.

Response: We thank the reviewer for these suggestions. Please see lines 369-389 and lines 433 – 441. 

Comment #56: P13. Compare your results of lumbar effects being better to those of Schreiber et al. I believe they did not find the same curve pattern as the one with the best response. Could it be that some benefits for your thoracic group may have been missed due to not using the chest twister and prone/supine exercises

Response: It is possible, we thank the reviewer for this valuable discussion point. I remember that there were conference papers, presented in SOSORT 2019 in SF, there were evidences showing that the spine responded better during exercise in prone. However, to our knowledge, this evidence was not published yet and unfortunately our study was initiated before 2019. Positions were designed to cope with patients’ daily activity as much as possible in this study. In Schreiber’s study, they used chest twist for thoracic curve, however, their results also showed the 3CP was significantly correlated with the largest Cobb after treatment, which was explained by the proven higher progression risk was shown in thoracic type. Therefore, it will be not suitable to compare our study with theirs on this point, because they also found the less effects with 3CP. Therefore, we alternatively stated that our study extended their results for the long term effect (lines 341-344) Moreover, for the potential variation caused by varied exercises, we further explained in lines 372-389, which was more emphasizing on the spinal biomechanics instead of varied positioning during exercise.

Comment #57: Please discuss the limitation of self-reporting compliance and possible recall issues depending on when your participants completed the compliance questionnaires.

Response: Participant need to take a note and sign on his checklist on the day when they performed exercise, then hand the checklist to physiotherapist at their next visit. See lines 139 – 143, recall effects were discussed on lines 405 - 408.

Comment #58: Limitations. There are more than 4 patterns in RIGO classification. Type 1 and type 2 were not discussed here.

Response: Information added. See lines 416 to 426.

Comment #59: Please discuss not monitoring co-interventions such as manipulations, massage, other fitness activities or self-used of off the shelve bracing.

Response: The participants would agree with only PSSE but no alternative exercise before sign the study consent form, but we agree that participants are likely to take another therapy without informing us. Commencing any other treatment were excluded from this study, please see line 115 and S2 Figure.

Comment #60: Could you review how many patients in each group experience a change of curve type over time and report which changes occurred.

Response: There were no curve pattern changed in this group of participants. But curve pattern might be changed with long-term exercise and should be identified. Please see Table 4

Comment #61: Discuss the imbalanced in the number of braced participant and possible brace compliance effects on results.

Response: Only 10 subjects had brace treatment and half of them only used night time brace. The logistic regression showed bracing did not affect our outcome, therefore, it maybe not very valuable to go in details of this analysis due to our small sample size. Please see lines 405 – 408. 

Comment #62: Discuss whether after 2 years of follow-up these participants had reached skeletal maturity. How many had reached discharge point or were sufficiently far after peak growth velocity to deem the results final or not.

Response: Fantastic point. Table 4 was edited to this purpose. 

Comment #63: The SRS-SOSORT recommendations propose a number of analysis reporting guidelines. Could you report Risser signs in your dataset or table 1. Could you report number of patients with curve over 30 at each time point. I believe none exceeded surgery threshold of 45 or 50 but this could be specified.

Response: Table 1 has been amended and see an updated table 4.

Comment #64: The discussion could compare how your follow-up length and results compared to other Schroth studies. Are your results better than others or similar. Was the progression risk of your cohort similar or worse than others.

Response: Fantastic point, related result and discussion were addressed accordingly. See lines 339 to 344. 

Comment #65: Please add a paragraph about whether there is a risk of overtreatment in this cohort. Could some of the patients have been left alone and avoid the burden of treatment. I can see that some would argue that for the fact that these patients possibly had no risk of progressing to bracing or surgery. However you showed some notable regression. This may appeal to some participants with small curves at low risk of progression. Still the possible overtreatment issue should be discussed.

Response: We thank the reviewer for this important point. The discussion was amended accordingly. See lines 353 to 356.

Comment #66: An important limitation in your study is that you may have been underpowered to detect difference between groups. It would be important to discuss how big a different in effects between curve types may be clinically important. This may be difficult to determine but a discussion of this topic should be presented.

Response: We agree. This has been added to line 417 to 425 and lines 446 to 449.

Comment #67: I would recommend against including the following in the discussion as you did not measure flexibility and did not study relation of effects of PSSE with baseline curve magnitude per say. However, further studies are necessary to address the correlation between spinal flexibility and the correction effects of PSSE at different curve magnitudes. IF you wish to keep then please move earlier in the discussion as suggestions for future research.

Response: We thank the reviewer for this comment. Please see lines 446 to 449. We re-centered our discussion point in the introduction and discussion, respectively.

Comment #68: I have annotated a few elements of the paper in an uploaded copy.

Response: We thank the reviewer for these comments. Thanks for nice editing, the format was amended accordingly.

Reviewer #3

Comment #1: The objective of this study is to investigate the comparative effectiveness of the PSSE correction effect on the Cobb angle between the thoracic and lumbar curves in AIS subjects. The authors considered a prospective clinical controlled study. While the study objectives sound interesting, a number of shotcomings were observed, in regards to abiding by the CONSORT guidelines for conducting and reporting results of high-quality randomized controlled trials (RCTs).

Response: We thank the reviewer for these comments.

Comment #2: 1. Abstract: The authors state results as: "A significant Cobb angle reduction was observed...", without any statement of p-values, estimated effect size, and its precision as confidence intervals, or CIs (say, 95\\%). This can appear confusing, and half-baked to a reader. Check CONSORT checklist for Abstracts reporting of RCTs, and rewrite the Abstract following guidelines.

Response: This was not an RCT study, therefore no needs for randomization etc, nonetheless, the problems in the abstract was recognized and amended.

Comment #3: 2. Methods:

Methods reporting appeared very messy. An orderly manner is suggested, following CONSORT guidelines, without repeating information, such as Trial Design, Participant Eligibility Crtieria and settings, Interventions, Outcomes, sample size/power considerations, Interim analysis and stopping rules, Randomization (details on random number generation, allocation concealment, implementation), Blinding issues, etc. The authors are advised to create separate subsections for each of the possible topics (whichever necessary), and that way produce a very clear writeup.

Response: Thanks for the valuable suggestions. We have revised the formatting.

Comment #4: (a) For instance, the randomization and allocation concealment should be made very clear; the trial staff recruiting patients should not have the randomization list. Randomization should be prepared by the trial statistician, and he/she would not participate in the recruiting. I am confused; was randomization not done during the "Allocation" phase in your CONSORT diagram?

Response: This was not an RCT, it was a CCT hence no randomization was required. Instead we controlled the curve pattern for the study.

Comment #5: (b) I am surprized to see no statement on sample size/power in a manuscript proposing a (clinical) controlled trial. This is really the key here!

Response: We thank the reviewer for the comment. The sample size estimation is in lines 192-196.

Comment #6: (c) t-tests were used for assessing group differences for continuous variables (under the assumption of Normality). What if underlying normality assumptions are violated? Why not non-parametric (robust) tests were proposed?

Response: Related clarification was addressed; normality was measured before commencing the analysis. Please see lines 195 - 213.

Comment #7: (d) Similarly, the one-way repeated measures ANOVA may also be replaced/presented by a nonparametric Friedman-type test. I mean, justification is needed for the underlying Normality assumptions.

Response: These have been clarified in the statistical analysis section.

Comment #8: 3. Results & Conclusions:

(a) The authors should check that any statement of significance should be followed by a p-value in the entire Results section.

Response: We thank the reviewer for this comment. This has been added.

Comment #9: (b) The authors admitted a long list of limitations in their work, notably, the "...unable to blind the subjects, or physiotherapists...". Despite the justification provided, I am not sure how good the trial is! With 40 subjects recruited, the results stated, at best, can only be claimed as from a pilot study. This needs to be clearly stated, and the study cannot be claimed as a nicely planned randomized trial. Also, they need to state that future studies (with larger sample sizes) are warranted to really understand the comparative efficiacy.

Response: We thank the reviewer for these comments. Please see the additions in lines 410-441.

---

## [Decision Letter · Decision Letter 1]

19 Oct 2020

PONE-D-19-35759R1

Dose curve pattern impact on the effects of physiotherapeutic scoliosis specific exercises on Cobb angles of participants with adolescent idiopathic scoliosis: a prospective clinical trial with two years follow-up

PLOS ONE

Dear Dr. Fan,

Thank you for submitting your manuscript to PLOS ONE. After careful consideration, we feel that it has merit but does not fully meet PLOS ONE’s publication criteria as it currently stands. Therefore, we invite you to submit a revised version of the manuscript that addresses the points raised during the review process.

ACADEMIC EDITOR:

Please consider carefully the remaining reviewers comments below. Whilst overall the judgement is minor amendments, there are some important points to still address.

We look forward to receiving your revised manuscript.

Kind regards,

Alison Rushton

Academic Editor

PLOS ONE

Reviewers' comments:

Reviewer's Responses to Questions

**Comments to the Author**

1. If the authors have adequately addressed your comments raised in a previous round of review and you feel that this manuscript is now acceptable for publication, you may indicate that here to bypass the “Comments to the Author” section, enter your conflict of interest statement in the “Confidential to Editor” section, and submit your "Accept" recommendation.

Reviewer #2: (No Response)

Reviewer #3: All comments have been addressed

2. Is the manuscript technically sound, and do the data support the conclusions?

Reviewer #2: Partly

Reviewer #3: (No Response)

3. Has the statistical analysis been performed appropriately and rigorously? 

Reviewer #2: No

Reviewer #3: (No Response)

4. Have the authors made all data underlying the findings in their manuscript fully available?

Reviewer #2: Yes

Reviewer #3: (No Response)

5. Is the manuscript presented in an intelligible fashion and written in standard English?

Reviewer #2: Yes

Reviewer #3: (No Response)

6. Review Comments to the Author

Reviewer #2: PONE – D -19- 35759R1

The Title should be corrected and show DOES not DOSE

Does curve pattern impact on the effects of physiotherapeutic scoliosis specific exercises on Cobb angles of participants with adolescent idiopathic scoliosis: a prospective clinical trial with two years follow-up

Specify in the abstract that the study is a non-randomized prospective trial.

In the abstract and the text line 203 The analysis should be announced as a mixed model anova (2 group by N time). Avoid stating 1 way as you have two factors. Group is a between groups factor and time is a within group factor.

RESPONSES to REVIEWER 1.

Your sentence on line 231 to 235 is unclear. There is an important difference between groups in the numbers of patients with brace. Specify that in the text. Also Did all braced patients have a Cheneau? the sentence suggests you only talk about those wearing a Cheneau brace but does not clearly state whether other types were used. Specify at what point correction was measured. Do you mean out of brace at the end of the study or in-brace and if yes then when was in-brace correction measured.

Abstract.

The following sentence in the abstract could be replaced by results of your regression. Your objective was focused on effect of curve type. This sentence does not add to this objective:

In addition, 65% of 37 participants (n=26) reached skeletal maturity (Risser sign = 5) and achieved significant reductions 38 in progression risk at the 2-year follow-up (Lonstein and Carlson Risk of progression: 0.45 ± 0.32 39 vs. 0.13 ± 0.14, p <0.001).

Obviously, patients reaching maturity would have significantly reduced progression risk. If you keep this idea in the abstract, please specify what this means with regards to their curve types or having done exercises.

In the consort flow chart please specify the number of cases that were excluded because they did not attend all the prescribed exercises sessions. (specify none did if that is the case.

Figure 2 f. The picture is stretched too wide. Please present in the original aspect ratio.

Page 13 line 203. Specify how you calculated D-value. Did you do for example 6 months minus baseline or basline minus 6 months. Specifying will help interpret mean d-values as indicating improved curves or deterioration.

P14 Line 206 Please refer to comment above about using mixed model anova and not one way ANOVA.

L218 It would be interesting to see the distribution of the hypermobility cases excluded in terms of their presenting a thoracic or lumbar dominant pattern given that you suspected a higher relation with single thoracic curves.

P14 Is single left lumbar atupical? Single right lumbar would be more atypical? Is this an error? Similarly, with single left thoracolumbar. Is this really atypical?

L229 to 231. I continue to disagree that you should compare baseline characteristics between groups statistically. Instead you should report what appears to be clinically important differences and maybe announced that this would be justification to explore these characteristics as covariables later. The study did not plan its sample size to detect such differences. The key is to explore if they play a role in confounding the results later.

L230 Please spell out what LCR stands for.

L231. The imbalance in bracing may not have reached significance but it should be pointed out.

L237 Table 1. If you keep the p values add which test, they came from in the table legend on in the column with the p-values. This would help the informed reader determine if your test likely had enough power.

L237 Here or in the methods state when the in-brace Cobb angle was measured.

L237 Refer to 1st year and 2nd year in the exercise compliance please.

L246 Since in the analysis section you left 2 choices for testing this different please report what test was used to compare the groups here with the p-value.

L248 Similarly. Please report the test used from comparisons within the thoracic group.

L267 Table 2. Instead of group labels of 0 and 1 can you spell thoracic and lumbar or use T and L for clarity?

L272 to 279. This analysis offers more information than that of the D-values. I would only report this ANOVA and possibly compared the frequencies of deterioration, stabilised or improved statistically as well. There is in fact a lot of redundancy between the analyses in table 2 and 3.

L289. It is less clear why this second case is highlighted. Only because of the large baseline curve? If yes specify that this case it highlighted because or the large curve. I was wondering if it was because it was the second case having refused the brace. Please clarify.

L309 To 311. It is redundant to repeat the baseline mean in each of the parentheses. Clarify which test produced the p-values. This set of results is surprising as it would suggest that the interaction should be significant the pattern of differences between timepoints was not the same between the two groups, yet you report that the interactions was not significant. Since the interaction was not significant you should not put emphasis on the different in changes over time between the groups.

L280 to 287. 312. To 317 Ideally the frequency results of the two groups would be reported together in terms of whether the distributions of improved , stable or deteriorated were similar or not between groups. Specify which test was used to test for those differences in the distributions.

L321 322. Focusing on a reduction in the progression risk is possibly misusing the formula of Lonstein and Carlson since it uses age and Risser so heavily and you did not report the formula in your paper. The change in angles would not play a clear role and this may mislead readers to think the reduction is due to the PSSE when maturity plays a big role. Their study possibly did not examine patients starting this late to really use the equation at this stage of treatment. If you keep this analysis, please report the full formula and add a discussion related to maturation in the discussion in relation to this reduced risk.

L328. Please add units for Age, the Cobb angle mean lines

L323 Your study likely did not have the power to examine so many variables in the logistic regression. Since your focus is on the curve type. Can you report the OR for the association between curve type and the 2 year curve outcomes? Once you enter multiple categorical predictors there really isn’t a lot of observations with the regression outcome to determine prediction estimates.

L351 to 353. You cannot claim you demonstrated Superiority of PSSE. You did not compare PSSE to an alternate intervention. You showed all patients receiving PSSE have avoided curve progression but not superiority to other treatments or to natural history (unless you refer to historical control and predictions given the progression risk.)

L359 to 361. Again, I disagree with emphasizing the change in progression risk. I do not believe the formula can be used this way. Further once again you omit to emphasize that much of this reduce regression risk is due to the change in age and Risser. It is misleading to imply the change is mostly due to the effect of PSSE and reduced Cobb angles. It is good indeed that curves did not progress, and patients are now close to maturity but here it would be more relevant to refer to natural history data showing how patients with certain curve magnitude at maturity progress into adulthood. EG.

L366 to 369 again you emphasise a different in the patterns of curve angles changes over time between groups when you ANOVA suggested that there was no interaction. There could be data distribution issues affecting the results here. Which ANOVA model did you end up using? Was your data meeting all other assumptions than sphericity?

L371 What do you refer to when stating “Progression value was comparable between groups at the study initiation”?

L372 373 Since you did not measure spinal flexibility please refer instead to “by the assumed more important flexibility of the lumbar spine than the thoracic spine.

L378 to 381. I would avoid suggesting that muscle actions have a bigger role for lumbar corrections and that only spiral breathing play a role in the thoracic regions. Modern Schroth instruction does not make assumptions about what muscles do or how corrections are done it simply offers a variety of cues and correction instructions and facilitation technique to get the desired movement without emphasis on how.

Maybe the key is that breathing is more related to thoracic ok. The axial elongation, side shift are done in both regions. The difficulty with breathing is likely relevant.

L405. Please cite Kuru et al for the effects of supervision vs not.

L417 There are more types in the whole BSPTS classification. Maybe instead rephrase to suggest that your sample included patients representing 5 different types from 8 in the BSPTS classification.

L435 Refer to hypothesized higher flexibility or previously shown higher flexibility. You did not show that in your study.

In this limitation section please clearly acknowledge that your regression is underpowered.

In the data supplementary file please replace the variable name by study id. Remove initials and replace by participant id numbers to fully comply with confidentiality of study data.

I have added edits in the text using the PDF comment function with strikethrough for deletions and insertions in blue.

Reviewer #3: (No Response)

7. PLOS authors have the option to publish the peer review history of their article (what does this mean?). If published, this will include your full peer review and any attached files.

Reviewer #2: **Yes: **Eric Parent

Reviewer #3: No

---

## [Author Response · Author response to Decision Letter 1]

3 Dec 2020

REPLY TO REVIEWERS

Re: Ms. No. PONE-D-19-35759 - "Does Curve Pattern Impact on the Effects of Physiotherapeutic Scoliosis Specific Exercises on Cobb angle of Adolescent Idiopathic Scoliosis: A prospective clinical controlled trial with two years follow-up”

Dear Editor Alison Rushton, 

The authors would like to thank you and the Reviewers for all of your time and effort devoted to the review of our aforementioned manuscript. The Reviewers’ comments were indeed extremely insightful and greatly appreciated. As such, the authors would like to take this opportunity to address each and every concern the Reviewers noted in their review of our submission. In addition, where appropriate, we have also revised our manuscript accordingly. Major changes are made in light gray.

We believe that the Reviewers’ comments have helped improve the quality of our manuscript. We hope that you and the Reviewers will find our revised work suitable for publication in PLOS One. 

Editor

Recommendation#1：If applicable, we recommend that you deposit your laboratory protocols in protocols.io to enhance the reproducibility of your results. Protocols.io assigns your protocol its own identifier (DOI) so that it can be cited independently in the future.

Response：Thanks for recommendation. This is a clinical trial without laboratory protocols, this trial was online registered and the study protocol was open access on www.chictr.org.cn, trial registration: ChiCTR1900028073.

Reviewer #2

Comment #1: The Title should be corrected and show DOES not DOSE

Does curve pattern impact on the effects of physiotherapeutic scoliosis specific exercises on Cobb angles of participants with adolescent idiopathic scoliosis: a prospective clinical trial with two years follow-up

Response：We apologize for the typo. The title has been corrected.

Comment #2: Specify in the abstract that the study is a non-randomized prospective trial.

Response：Study design is specified in line 17.

Comment #3: In the abstract and the text line 203 The analysis should be announced as a mixed model anova (2 group by N time). Avoid stating 1 way as you have two factors. Group is a between groups factor and time is a within group factor.

Response：Analysis method was renamed accordingly, please see line 24-25.

Comment #4：Your sentence on line 231 to 235 is unclear. There is an important difference between groups in the numbers of patients with brace. Specify that in the text. 

Response：This was specified on lines 241 to 244.

Comments #5: Also did all braced patients have a Cheneau? the sentence suggests you only talk about those wearing a Cheneau brace but does not clearly state whether other types were used. Specify at what point correction was measured. Do you mean out of brace at the end of the study or in-brace and if yes then when was in-brace correction measured.

Response: 

1. All braced subjects were wearing Cheneau brace, sorry for this confusion, clarification was made on lines 237 to 239.

2.The in-brace correction was the initial in-brace correction (a percentage value of Cobb reduction during an X-ray with the brace fitted on the participant more than two hours), which was measured after a brace adaption period (two weeks). Please see lines 136 to 138 and lines 188 to 190. 

3.Regarding the Cobb angle at the end of the study, it was measured without brace and participants were required off-bracing at night prior to X-ray measurement in the next day. Please see lines 190 to 191.

Comment #6: Abstract. The following sentence in the abstract could be replaced by results of your regression. Your objective was focused on effect of curve type. This sentence does not add to this objective: In addition, 65% of 37 participants (n=26) reached skeletal maturity (Risser sign = 5) and achieved significant reductions 38 in progression risk at the 2-year follow-up (Lonstein and Carlson Risk of progression: 0.45 ± 0.32 39 vs. 0.13 ± 0.14, p <0.001).

Obviously, patients reaching maturity would have significantly reduced progression risk. If you keep this idea in the abstract, please specify what this means with regards to their curve types or having done exercises.

Response: Fantastic suggestion. We re-structured the result of abstract and emphasized result of correlation of curve pattern with curve regression. Please see lines 36 to 38.

Comment #7: In the consort flow chart please specify the number of cases that were excluded because they did not attend all the prescribed exercises sessions. (specify none did if that is the case)

Response: All participants completed prescribed exercise sessions. The clarification was further edited in S2 Fig. CONSORT flow diagram and line 127. 

Comment #8: Figure 2 f. The picture is stretched too wide. Please present in the original aspect ratio.

Response: The picture (Fig.2) is updated with original aspect ratio. Please check line 168. 

Comment #9: Page 13 line 203. Specify how you calculated D-value. Did you do for example 6 months minus baseline or basline minus 6 months. Specifying will help interpret mean d-values as indicating improved curves or deterioration.

Response: Thanks for this valuable suggestion. Specifying was made on lines 206 to 209. 

Comment #10: P14 Line 206 Please refer to comment above about using mixed model anova and not one way ANOVA.

Response: Thanks for this reminding and correction was addressed on line 211.

Comment #11: L218 It would be interesting to see the distribution of the hypermobility cases excluded in terms of their presenting a thoracic or lumbar dominant pattern given that you suspected a higher relation with single thoracic curves.

Response: The distribution of noted hypermobility in this study was reported on lines 224 to 225. We did not reproduce the same distribution as the previous study, yet we did find a few of patients presenting hypermobility regarding the Beighton score. The distribution of hypermobility in AIS can be explained better with a population level, cross-sectional study. 

Comment #12: P14 Is single left lumbar atupical? Single right lumbar would be more atypical? Is this an error? Similarly, with single left thoracolumbar. Is this really atypical?

Response: This is a really good question. Proven atypical curve patterns include a single left thoracic curve, a curvature with a loss of kyphosis, a rapid progression and ≤ 6 segmental scoliosis, etc. The single right lumbar and a left thoracolumbar were reported as atypical pattern in previous studies and in which subjects were found having an underlying syrinx presented with a MRI assessment. Therefore, cases with aforementioned curve pattern were excluded to control study heterogeneity yet prescribed with either a neurological assessment or an MRI assessment before commencing any treatments. We cited the related studies, please check lines 225 to 227 and references 35 to 38. 

Comment #13: L229 to 231. I continue to disagree that you should compare baseline characteristics between groups statistically. Instead you should report what appears to be clinically important differences and maybe announced that this would be justification to explore these characteristics as covariables later. The study did not plan its sample size to detect such differences. The key is to explore if they play a role in confounding the results later.

Response: Sure, we do agree that defining confounding factors is way more important than comparing the baseline statistically in a clinical trial, Thus, baseline characteristics were reported by descriptive statistics in this updated manuscript. Please check Table 1. In addition, bracing was set as a covariate with repeated measures to evaluate effects of PSSE in reducing Cobb angles. 

Comment #14: L230 Please spell out what LCR stands for.

Response: Lonstein Carlson Risk of progression (LCR-value) was firstly explained on the line 205, then the abbreviated term, LCR-value was adopted throughout manuscript.

Comment #15: L231. The imbalance in bracing may not have reached significance but it should be pointed out.

Response: Sure indeed, this was emphasized on the lines 241 to 244.

Comments #16: L237 Table 1. If you keep the p values add which test, they came from in the table legend on in the column with the p-values. This would help the informed reader determine if your test likely had enough power.

Response: We used descriptive statistics for baseline characteristics in this updated manuscript, thus no p value was reported in the table 1, but we adopted t-test and paired t-test to compare the exercise compliance of each year, which would help us on discussing how did exercise compliance affect the outcome. Thus, a table legend was made accordingly on the lines 250 to 252. 

Comment #17: L237 Here or in the methods state when the in-brace Cobb angle was measured.

Response: The timepoint of measuring in-brace Cobb angle was stated on the lines 136 to 138.

Comment #18: L237 Refer to 1st year and 2nd year in the exercise compliance please.

Response: Done. Please check the Table 1. 

Comment #19: L246 Since in the analysis section you left 2 choices for testing this different please report what test was used to compare the groups here with the p-value.

Response: This is a subgroup analysis to detect any changes of exercise compliance between groups on the 1st and 2nd year follow-up. Hence, an independent t-test, and paired t-test were used to compare the exercise compliance between and within groups. Please check lines 256 to 257.

Comment #20: L248 Similarly. Please report the test used from comparisons within the thoracic group.

Response: This is a subgroup analysis to detect any changes of exercise compliance between groups on the 1st and 2nd year follow-up. Hence, an independent t-test, and paired t-test were used to compare the exercise compliance between and within groups. Please check the lines 256 to 257.

Comment #21: L267 Table 2. Instead of group labels of 0 and 1 can you spell thoracic and lumbar or use T and L for clarity?

Response: Thanks for this suggestion. Revision was made on the Table 2.

Comment #22: L272 to 279. This analysis offers more information than that of the D-values. I would only report this ANOVA and possibly compared the frequencies of deterioration, stabilized or improved statistically as well. There is in fact a lot of redundancy between the analyses in table 2 and 3.

Response: Agree. Thanks for this fantastic suggestion to make our points much clearer. Hence, we withdrew the previous table 2 (repeated measures of D-value) and re-centered our primary analysis on the raw Cobb angles. Please see lines 269 to 286, and the updated table 2. 

Comment #23: L289. It is less clear why this second case is highlighted. Only because of the large baseline curve? If yes specify that this case it highlighted because or the large curve. I was wondering if it was because it was the second case having refused the brace. Please clarify.

Response: The second case refused the brace. The clarification was made on the lines 320 - 321.

Comment #24: L309 To 311. It is redundant to repeat the baseline mean in each of the parentheses. Clarify which test produced the p-values. This set of results is surprising as it would suggest that the interaction should be significant the pattern of differences between timepoints was not the same between the two groups, yet you report that the interactions was not significant. Since the interaction was not significant you should not put emphasis on the different in changes over time between the groups.

Response: Thanks for this comment, the redundancy was deleted. Please check the re-structured paragraphs on lines 294 to 321, using descriptive statistics to illustrate the trend of scoliosis regression and stabilization in each group. The repeated measures with Bonferroni adjustment showed a significant time-effect in reducing Cobb angles (raw value) over four testing times. Additionally, post hoc analysis showed the significant reductions of Cobb were occurred on the 1.5- and 2-year follow-up, respectively. Therefore, we used descriptive statistics, additionally to illustrate how many participants in which group showed clinical curve regression. In case of we withdrew the repeated measures of D-value of Cobb angles, hence the previous redundancy was completely deleted. 

Comment #25: L280 to 287. 312. To 317 Ideally the frequency results of the two groups would be reported together in terms of whether the distributions of improved, stable or deteriorated were similar or not between groups. Specify which test was used to test for those differences in the distributions.

Response: Fantastic suggestion, please see the re-structured paragraphs on lines 294 to 321. The distribution of scoliosis progression, stabilization and regression was analysed using Chi-square analysis. Please see lines 296 to 298.

Comment #26: L321 322. Focusing on a reduction in the progression risk is possibly misusing the formula of Lonstein and Carlson since it uses age and Risser so heavily and you did not report the formula in your paper. The change in angles would not play a clear role and this may mislead readers to think the reduction is due to the PSSE when maturity plays a big role. Their study possibly did not examine patients starting this late to really use the equation at this stage of treatment. If you keep this analysis, please report the full formula and add a discussion related to maturation in the discussion in relation to this reduced risk.

Response: Fantastic point, thanks for pointing it out, we realized that comparing LCR-value after two years follow-up was inappropriate. Therefore, we withdrew the comparison of LCR-value at the 2-year follow-up and replaced it to a description of how much participants reached skeletal maturity (Risser 5) and presented with a curve less than 30 degrees. Please see lines 294 to 295 and Table 3.

Comment #27: L328. Please add units for Age, the Cobb angle mean lines

Response: Done, Thanks for this suggestion. Please see line 323 (Table 3).

Comment #28: L323 Your study likely did not have the power to examine so many variables in the logistic regression. Since your focus is on the curve type. Can you report the OR for the association between curve type and the 2 year curve outcomes? Once you enter multiple categorical predictors there really isn’t a lot of observations with the regression outcome to determine prediction estimates.

Response: Nice point, thanks for pointing it out, we now only enrolled curve pattern and bracing (since we enrolled bracing into repeated measures) into the logistic regression model to further illustrate the effects of bracing and curve pattern in scoliosis regression with PSSE. Please see lines 299 to 302. However, this logistic regression was underpowered since this study was not originally designed for a regression analysis, thus, we acknowledged this limitation on the lines 449 to 454. 

Comment #29: L351 to 353. You cannot claim you demonstrated Superiority of PSSE. You did not compare PSSE to an alternate intervention. You showed all patients receiving PSSE have avoided curve progression but not superiority to other treatments or to natural history (unless you refer to historical control and predictions given the progression risk.)

Response: Agree, we replaced “superiority of PSSE” by “effectiveness of PSSE” in preventing scoliosis progression. Please see lines 358 to 359.

Comment #30: L359 to 361. Again, I disagree with emphasizing the change in progression risk. I do not believe the formula can be used this way. Further once again you omit to emphasize that much of this reduce regression risk is due to the change in age and Risser. It is misleading to imply the change is mostly due to the effect of PSSE and reduced Cobb angles. It is good indeed that curves did not progress, and patients are now close to maturity but here it would be more relevant to refer to natural history data showing how patients with certain curve magnitude at maturity progress into adulthood. EG.

Response: Fantastic point, thanks for pointing it out, we realized that comparing LCR-value after two years follow-up was inappropriate. Therefore, we withdrew the comparison of LCR-value at the 2-year follow-up and replaced it to a description of how much participants reached skeletal maturity (Risser 5) and presented with a curve less than 30 degrees. Please see lines 366 to 368 and Table 3.

Comment #31: L366 to 369 again you emphasise a different in the patterns of curve angles changes over time between groups when you ANOVA suggested that there was no interaction. There could be data distribution issues affecting the results here. Which ANOVA model did you end up using? Was your data meeting all other assumptions than sphericity?

Response: Sorry for this confusion, agree that if there were no interaction effects in reducing Cobb angles, it was inappropriate to further compare variables between groups. In this study, the significance was only detected in the within-subjects comparison, having a time effect in reducing Cobb angles for all participants (observed power 0.7, lines 275 - 278). A separate repeated analysis with a Bonferroni adjustment (control type I error) further revealed that the significant reduction of Cobb angles was dominant in the lumbar group (observed power 0.95, lines 282 - 283). Therefore, we rewrote this part (please see lines 371 to 372). Levene’s test indicated that Cobb angles at the baseline met homogeneity of variance assumption (p=0.5). Mauchly test indicated that the assumption of sphericity was violated (χ2 =32.1 , df =5 , p < 0.001), hence, the degrees of freedom were corrected using Greenhouse-Geisser estimates of sphericity. Please see lines 271 to 276. The assumption of Greenhouse-Geisser was met with an Epsilon value 0.72. 

Comment #32: L371 What do you refer to when stating “Progression value was comparable between groups at the study initiation”?

Response: Sorry for the confusion, the comparable here means the proportion was similar between groups, the clarification was made on lines 376 to 378. 

Comment #33: L372 373 Since you did not measure spinal flexibility please refer instead to “by the assumed more important flexibility of the lumbar spine than the thoracic spine.

Response: Fantastic points. Please see the reediting on lines 378 to 379.

Comment #34: L378 to 381. I would avoid suggesting that muscle actions have a bigger role for lumbar corrections and that only spiral breathing play a role in the thoracic regions. Modern Schroth instruction does not make assumptions about what muscles do or how corrections are done it simply offers a variety of cues and correction instructions and facilitation technique to get the desired movement without emphasis on how.Maybe the key is that breathing is more related to thoracic ok. The axial elongation, side shift are done in both regions. The difficulty with breathing is likely relevant.

Response: Nice points, thanks for this explanation, the revised discussion was made on the lines 385 to 387.

Comment #35: L405. Please cite Kuru et al for the effects of supervision vs not.

Response: cited. Thanks for this recommendation. Please check reference 49.

Comment #36: L417 There are more types in the whole BSPTS classification. Maybe instead rephrase to suggest that your sample included patients representing 5 different types from 8 in the BSPTS classification.

Response: Nice points, thanks for this explanation, the revised discussion was made on lines 423-424.

Comment #37: L435 Refer to hypothesized higher flexibility or previously shown higher flexibility. You did not show that in your study.

Response: Thanks for this reminding, we acknowledge this limitation on lines 444 to 445. 

Comment #38: In this limitation section please clearly acknowledge that your regression is underpowered.

Response: Indeed. The acknowledgement was made on lines 448 to 454. 

Comment #39: In the data supplementary file please replace the variable name by study id. Remove initials and replace by participant id numbers to fully comply with confidentiality of study data.

Response: Done, thanks for this important recommendation. Please check the updated S4 File. Study data.

Comment #39: I have added edits in the text using the PDF comment function with strikethrough for deletions and insertions in blue.

Response: We have revised the format accordingly. Deeply appreciate your detailed comments.

---

## [Decision Letter · Decision Letter 2]

22 Dec 2020

PONE-D-19-35759R2

Does curve pattern impact on the effects of physiotherapeutic scoliosis specific exercises on Cobb angles of participants with adolescent idiopathic scoliosis: a prospective clinical trial with two years follow-up

PLOS ONE

Dear Dr. Fan,

Thank you for submitting your manuscript to PLOS ONE. After careful consideration, we feel that it has merit but does not fully meet PLOS ONE’s publication criteria as it currently stands. Therefore, we invite you to submit a revised version of the manuscript that addresses the points raised during the review process.

ACADEMIC EDITOR:

Please address the minor points made by reviewer #2.

We look forward to receiving your revised manuscript.

Kind regards,

Alison Rushton

Academic Editor

PLOS ONE

Reviewers' comments:

Reviewer's Responses to Questions

**Comments to the Author**

1. If the authors have adequately addressed your comments raised in a previous round of review and you feel that this manuscript is now acceptable for publication, you may indicate that here to bypass the “Comments to the Author” section, enter your conflict of interest statement in the “Confidential to Editor” section, and submit your "Accept" recommendation.

Reviewer #2: All comments have been addressed

2. Is the manuscript technically sound, and do the data support the conclusions?

Reviewer #2: Yes

3. Has the statistical analysis been performed appropriately and rigorously? 

Reviewer #2: Yes

4. Have the authors made all data underlying the findings in their manuscript fully available?

Reviewer #2: (No Response)

5. Is the manuscript presented in an intelligible fashion and written in standard English?

Reviewer #2: Yes

6. Review Comments to the Author

Reviewer #2: 

Thanks for your thorough consideration of my suggestions and revisions to the manuscript.

L114. Selection criteria was complete attendance at physiotherapy sessions. This is problematic. Ideally a per protocol and an intent-to-treat analysis would be reported. Also it will be important to report how common it was to exclude participants in each group for this reason. Please specify in figure 1 is any participants were excluded (could not start the study before division into groups) for not completing all exercise sessions.

L213. Please specify the specify pairwise comparison test used in the effect of significant ANOVA main effects. You do so later in the results but it should appear here (Bonferonni)

See minor English edits in the file annotated.

7. PLOS authors have the option to publish the peer review history of their article (what does this mean?). If published, this will include your full peer review and any attached files.

Reviewer #2: **Yes: **Eric C. Parent

---

## [Author Response · Author response to Decision Letter 2]

27 Dec 2020

Re: Ms. No. PONE-D-19-35759R2 - "Does Curve Pattern Impact on the Effects of Physiotherapeutic Scoliosis Specific Exercises on Cobb angle of Adolescent Idiopathic Scoliosis: A prospective clinical controlled trial with two years follow-up”

Dear Editor Alison Rushton, 

The authors would like to thank you and the reviewer, Eric Parent for all of your time and effort devoted to the review of our aforementioned manuscript. The reviewer’ comments were indeed extremely insightful and greatly appreciated. As such, the authors would like to take this opportunity to address each and every concern the reviewer noted in his review of our submission. In addition, where appropriate, we have also revised our manuscript accordingly. Minor changes are made in the revised manuscript with track changes.

We believe that the reviewer, Eric Parent’s comments have helped improve the quality of our manuscript. We hope that you and the Reviewers will find our revised work suitable for publication in PLOS One.

Academic editor Alison Rushton

Comment #1: Please address the minor points made by reviewer #2

 Response：All minor reversions were addressed accordingly. 

Reviewer #2

Comment #1: Selection criteria was complete attendance at physiotherapy sessions. This is problematic. Ideally a per protocol and an intent-to-treat analysis would be reported. Also it will be important to report how common it was to exclude participants in each group for this reason. Please specify in figure 1 is any participants were excluded (could not start the study before division into groups) for not completing all exercise sessions. 

Response：Sorry for this confusion made in the last manuscript. The “all exercise sessions” refers to the once per month of SUPERVISED PSSE sessions within the first 6 months. This was to review the performed exercise monthly on the early phase of this study to control learning effects. Clarification was made in this revised manuscript on lines 116 to 117 and S2 Fig. 1. 

 The Intention to treat (ITT) analysis and per-protocol analysis were reported on lines 212 to 215. However, ITT analysis was same as the per-protocol analysis in this study because there were no dropouts or crossovers during the whole study period, the final analysis was performed based on their original allocation and all participants attended supervised PSSE and reported their home exercise compliance periodically, thus, per-protocol analysis was performed instead of ITT to avoid redundancy, please check S2 Fig.1. 

Comment #2: Please specify the specify pairwise comparison test used in the effect of significant ANOVA main effects. You do so later in the results but it should appear here (Bonferonni).

Response：Repeated measures with the Bonferroni adjustments were adopted for post-hoc pairwise comparisons. This specification was made on lines 217 to 218. 

Comment #3：See minor English edits in the file annotated. 

Response: We have revised the manuscript accordingly. Deeply appreciate your time and valuable comments.

---

## [Editor Report · Decision Letter 3]

11 Jan 2021

Does curve pattern impact on the effects of physiotherapeutic scoliosis specific exercises on Cobb angles of participants with adolescent idiopathic scoliosis: a prospective clinical trial with two years follow-up

PONE-D-19-35759R3

Dear Dr. Cheung,

We’re pleased to inform you that your manuscript has been judged scientifically suitable for publication and will be formally accepted for publication once it meets all outstanding technical requirements.

Kind regards,

Alison Rushton

Academic Editor

PLOS ONE

Additional Editor Comments (optional):

Thank you for addressing all of the reviewers' comments to improve the quality of the manuscript..

---

## [Editor Report · Acceptance letter]

13 Jan 2021

PONE-D-19-35759R3 

Does curve pattern impact on the effects of physiotherapeutic scoliosis specific exercises on Cobb angles of participants with adolescent idiopathic scoliosis: a prospective clinical trial with two years follow-up 

Dear Dr. Cheung:

I'm pleased to inform you that your manuscript has been deemed suitable for publication in PLOS ONE. Congratulations! Your manuscript is now with our production department. 

Kind regards, 

on behalf of

Professor Alison Rushton 

Academic Editor

PLOS ONE